# GEDAN: Learning the Edit Costs for Graph Edit Distance

## Abstract

Graph Edit Distance (GED) is defined as the minimum cost transformation of one graph into another and is a widely adopted metric for measuring the dissimilarity between graphs. The major problem of GED is that its computation is NP-hard, which has in turn led to the development of various approximation methods, including approaches based on neural networks (NN). However, most NN methods assume a unit cost for edit operations – a restrictive and often unrealistic simplification, since topological and functional distances rarely coincide in real-world data. In this paper, we propose a fully end-to-end Graph Neural Network framework for learning the edit costs for GED, at a fine-grained level, aligning topological and task-specific similarity. Our method combines an unsupervised self-organizing mechanism for GED approximation with a Generalized Additive Model that flexibly learns contextualized edit costs. Experiments demonstrate that our approach overcomes the limitations of non–end-to-end methods, yielding directly interpretable graph matchings, uncovering meaningful structures in complex graphs, and showing strong applicability to domains such as molecular analysis.

## 1 Introduction and Related Work

Graphs are a versatile data structure that can be used to represent both the features of entities and the relationships between them. Due to their power and flexibility, graphs have found numerous applications. For example, graph-based representations are successfully used in the analysis of social networks (Chen, 2013), the recognition of human poses (Mathis et al., 2018) or the representation of water bodies (Fankhauser et al., 2024). Graphs can also be used in bio-informatics and computational chemistry, for example, to represent proteins (Bernstein et al., 1977) or molecules (Bento et al., 2020), the field we consider as possible application in this paper.

In the last decades, several methods have been proposed and researched to assess the dissimilarity or similarity between graphs, such as diverse graph kernels or graph matching procedures (Conte et al., 2004; Vento, 2013). Among these, *Graph Edit Distance* (GED) (Bunke & Allermann, 1983) can be interpreted as a standard approach. Basically, GED measures the distance between two graphs in terms of the minimum cost operations required to transform one graph into the other. The considered operations typically include insertions, deletions, and substitutions of both nodes and edges. One of the most important advantages of GED is that it can be applied to virtually all types of graphs and thus, GED can be flexibly adapted to any graph-based application. However, exact computation of GED is an NP-hard problem (Bunke, 1997), which makes its application to larger graphs, or in scenarios requiring a large number of comparisons, impractical.

To address this limitation and to reduce the computational complexity of GED, several strategies for relaxing the problem have been proposed. These techniques range from binary linear programming (Justice & III, 2006), over fast suboptimal algorithms preserving classification accuracy (Neuhaus et al., 2006b; Riesen & Bunke, 2009b), to quadratic assignment models (Bougleux et al., 2017) and local search algorithms (Boria et al., 2020). More recently, diverse approaches to learn and approximate GED using *Graph Neural Networks* (GNNs) have been proposed (Bai et al., 2019a; Ranjan et al., 2022; Zhuo & Tan, 2022). A very recent approach (Jain et al., 2024), also based on GNNs, differs from the others in explicitly formulating the GED as a matrix optimization problem. That is, this method approximates the GED by learning the permutation that minimizes the assignment cost between the substructures of two graphs.

Most existing work approaches GED primarily as an approximation problem for measuring graph dissimilarity, typically assuming fixed – often uniform – edit costs. This simplification is unrealistic, since treating all edits as equally costly ignores the fact that structural similarity is not necessarily aligned with functional or task-specific similarity. In molecular graphs, for example, the substitution of a single atom (e.g., replacing a hydrogen with a chlorine) can drastically alter pharmacological activity, toxicity, or binding affinity. Conversely, two molecules that are topologically quite dissimilar may nonetheless exhibit comparable biological properties – a phenomenon known as scaffold hopping in drug design (Sun et al., 2012).

Consequently, adapting GED to reflect meaningful, task-specific edit costs is non-trivial, since most available methods are supervised and rely on pre-computed distances. To make GED reflect functional similarity, it is necessary to learn the underlying edit costs – quantities that are not known a priori but constitute the very target of learning. While unsupervised methods such as GraphCL (You et al., 2020) or UGraphEmb (Bai et al., 2019b) can learn structural representations through augmentation, they cannot explicitly capture the actual edit costs. The problem of learning edit costs has a long history: early work pursued probabilistic and statistical formulations (Neuhaus & Bunke, 2004; 2007), while subsequent approaches relied on optimization strategies that are not fully differentiable end-to-end (Cortés et al., 2019; Garcia-Hernandez et al., 2020; Rica et al., 2021). More recent methods such as EPIC (Heo et al., 2024) and GMSM (Pellizzoni et al., 2024) introduce differentiable pipelines and attempt to learn edit costs. EPIC employs them primarily for data augmentation, while GMSM restricts node-specific costs to the final matching step, thereby missing multi-scale granularity.

In this paper, we address the problem of directly learning edit costs that enable GED to align with task-relevant properties. Building on the approach of Jain et al. (2024), we combine GNNs with *Generalized Additive Models* (GAM) (McCaffrey, 1992), and employ a Gumbel-Sinkhorn network (Mena et al., 2018) for soft GED approximation. This approximation is performed in an unsupervised, self-organizing manner driving the model to restructure itself to minimize the overall assignment cost. Finally, we apply contrastive learning to align structural and functional distances between graphs, thereby improving this alignment while providing interpretable insights into graph relationships.

The remaining paper is structured as follows. In Section 2, we describe both the method used to approximate the GED and the method to learn the costs of the edit operations. In Section 3, we present the results of our experimental evaluation and present a use case of our novel approach stemming from the field of computational chemistry. In Section 4, we conclude the paper by discussing its limitations and outlining directions for future research.

## 2 METHODS

In the present paper, we define a graph as a triple $G = (V, E, \mu)$, where $V$ is the set of nodes, $E \subseteq \{(u, v) | u, v \in V, u \neq v\}$ is the set of edges, and $\mu : V \to L$ is a function that assigns a label $\mu(v) = l \in L$ to each node $v \in V$ (where $L$ is a given label alphabet). Hence, we focus on node-labeled graphs, though our approach can be extended to include edge labels.

Our goal is not merely to compute the GED, but to learn a set of edit costs $\mathcal{C}$ such that GED aligns with a task-specific functional distance $d_f$. Formally, given three graphs $G, G', G''$, we aim to enforce

$$d_f(G, G') < d_f(G, G'') \implies GED(G, G'; \mathcal{C}) < GED(G, G''; \mathcal{C}), \tag{1}$$

ensuring that as functional distance increases, so does the corresponding GED. Consequently, the set of edit costs $\mathcal{C}$ is not fixed, but learned to preserve the ordering induced by $d_f$.

To approximate GED, our approach is based on a reformulation of the problem of GED to a *Quadratic Assignment Problem* (QAP) (Bougleux et al., 2017) as proposed in Jain et al. (2024). However, the method proposed in this paper differs from prior work in four key aspects, which we detail in the following subsections.

1. By adopting the message passing mechanism of GNNs, we introduce a system of additive penalties for dissimilar node representations. This means that we not only consider the nodes and their representations, but also take their growing neighboring structures into account.

2. Instead of learning the optimal permutation for the underlying QAP directly, we use a Gumbel-Sinkhorn network in a pre-trained way to approximate a *Linear Sum Assignment*

*Problem* (LSAP). Moreover, we define the framework so that it allows for unsupervised training, which enhances its practical applicability.

3. Our method learns the edit costs for nodes and edges and, to enable greater flexibility, we generalize these scalar costs to node-dependent cost matrices, which are learned via an auxiliary neural network.

4. The new formulation allows the model to assign context-aware, node-specific edit costs during training, exploiting a double loss deployed in other metric learning frameworks.

## 2.1 PENALIZATION WITH MESSAGE PASSING

Let $G = (V, E, \mu)$ and $G' = (V', E', \mu')$ be two graphs with $n$ and $m$ nodes, respectively. To make $G$ and $G'$ of equal size, we pad them with $m$ and $n$ dummy nodes termed $\epsilon$, which are all assigned a special label $\mu(\epsilon)$. Let $\tilde{V} = V \cup \{\epsilon\}_m$ and $\tilde{V}' = V' \cup \{\epsilon\}_n$ be the sets of nodes of the padded graphs, for any node $v \in \tilde{V}$ (or $v' \in \tilde{V}'$), we define the $k$-th level representation as

$$h^{(k)}(v) = \begin{cases} \mu(v), & \text{if } k = 0, \\ A^k\left(h^{(k-1)}(v)\right) + h^{(k-1)}(v), & \text{if } k > 0, \end{cases} \tag{2}$$

where $A^k$ is an injective neighborhood aggregation function that captures information up to the $k$-hop neighborhood of node $v$, while preserving its structural uniqueness. In our implementation, we instantiate $A^k$ as the *Graph Isomorphism Network* (GIN) (Xu et al., 2019), which generalizes the *Weisfeiler-Lehman* test (Shervashidze et al., 2011) and is known for its expressive power in distinguishing graph structures.

Next, we define $d^{(k)} : \tilde{V} \times \tilde{V}' \to \mathbb{R}_{\geq 0}$ as a distance function between $d$-dimensional node embeddings $h^{(k)}(v)$ and $h^{(k)}(v')$ at level $k$ by means of the normalized cosine dissimilarity

$$d^{(k)}(v, v') = \frac{1}{2}\left(1 - \frac{\langle h^{(k)}(v), h^{(k)}(v')\rangle}{\|h^{(k)}(v)\| \cdot \|h^{(k)}(v')\|}\right). \tag{3}$$

A distance $d^k(v, v')$ of 0 indicates identical representations $h^{(k)}(v)$ and $h^{(k)}(v')$, and values close to 1 indicate a high dissimilarity between these representations. We define the multi-scale distance $d^K(v, v')$ between nodes $v \in \tilde{V}$ and $v' \in \tilde{V}'$ as

$$d^K(v, v') = \sum_{k=0}^{K} d^{(k)}(v, v'). \tag{4}$$

This formulation enables a hierarchical comparison of nodes, where each term $d^{(k)}(v, v')$ evaluates the dissimilarity based on neighborhood information of $v$ and $v'$ of increasing radius. The resulting distance $d^K(v, v')$ is zero if, and only if, both nodes and their surrounding structures are equal.

As with GIN and its known limitations relative to the *Weisfeiler-Lehman* test, our method shares the same constraint: if two graphs are indistinguishable by GIN at depth $K$, they will also be indistinguishable by our architecture.

## 2.2 LEARNING THE MATCHING

In our model the matching between two graphs $G$ and $G'$, containing $n$ and $m$ nodes respectively, is learned by constructing a series of matrices of size $[n + m] \times [m + n]$, as illustrated in Figure 1 (a). A formal definition of these matrices is provided in Appendix B.

Each element $m_{ij}$ of the matrix $\mathbf{M}_k = (m_{ij})$ encodes the dissimilarity $d^{(k)}(v_i, v_j)$ between the representations learned by the GIN for node $v_i \in G$ and node $v_j \in G'$ at level $k$. The $K$ multi-scale matrices $\mathbf{M}_1, \ldots, \mathbf{M}_K$ are combined using the *Generalized Additive Model* (GAM) framework (Mc-Caffrey, 1992), which has recently been shown to improve interpretability in deep models (Bechler-Speicher et al., 2024). This mechanism allows penalizing mismatches, as illustrated in Figure 1 (b).

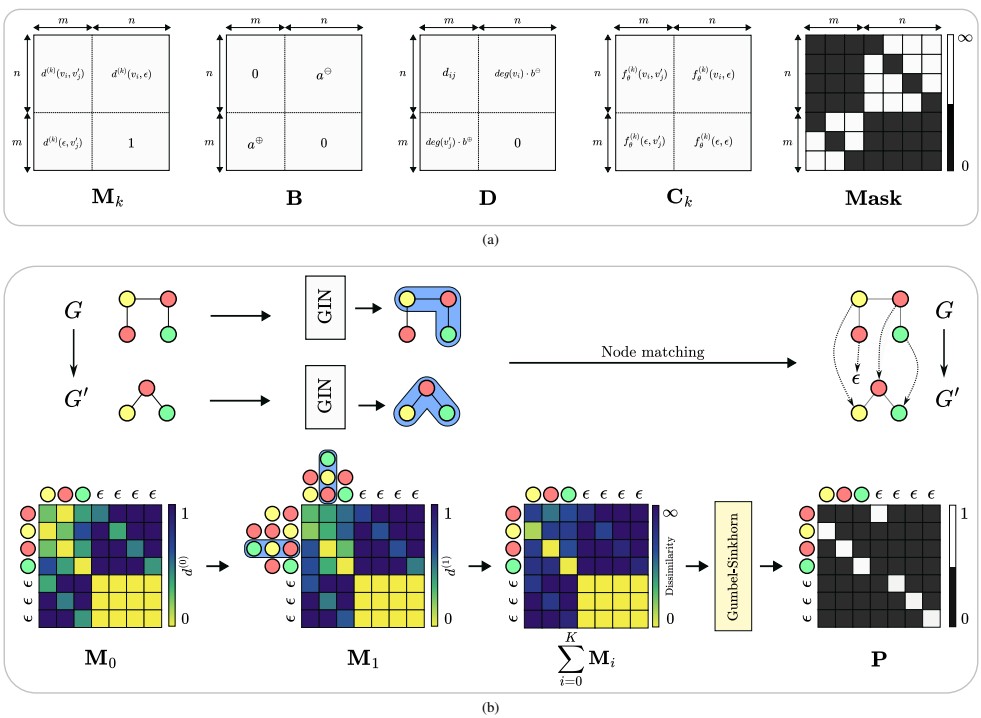

Figure 1: (a) shows the structure of the five matrices used in GEDAN. (b) is an example of how the distance $d^{(k)}$ penalizes mismatched nodes, allowing $\mathbf{P}$ to more precisely identify graph correspondences.

The entries $b_{ij}$ of matrix $\mathbf{B} = (b_{ij})$ encode insertion and deletion costs. Formally, $b_{ij} = a^\ominus$ if $i < n$ and $j \geq m$, $b_{ij} = a^\oplus$ if $i \geq n$ and $j < m$, and $b_{ij} = 0$ otherwise. That is, matrix $\mathbf{B}$ encodes these operations in its upper-right (deletions) and lower-left (insertions) blocks, with all other entries set to zero.

Matrix $\mathbf{D} = (d_{ij})$ stores the costs of the edge operations, which are implied by the corresponding node operations (assuming unlabeled edges), with

$$d_{ij} = \mathrm{ReLU}(\deg(v_i) - \deg(v'_j)) \cdot b^\ominus + \mathrm{ReLU}(\deg(v'_j) - \deg(v_i)) \cdot b^\oplus, \tag{5}$$

where $\deg(v_i)$ and $\deg(v'_j)$ denote the degrees of node $v_i \in G$ and node $v'_j \in G'$, respectively, and $b^\ominus$ and $b^\oplus$ correspond to the costs of edge deletions and insertions, respectively.

The sum $\sum_{k=0}^{K} \mathbf{M}_k + \mathbf{D} + \mathbf{B}$ aligns with the LSAP-based reformulation of GED (Riesen & Bunke, 2009b). In our formulation, however, we do not constrain insertions and deletions to diagonal entries; instead, a masking mechanism hides off-diagonal elements from the network to prevent them from affecting learning (see $\mathbf{Mask}$ in Figure 1 (a)).

Finally, using a pre-trained Gumbel-Sinkhorn network (Mena et al., 2018) to approximate LSAP, we calculate a soft permutation matrix $\mathbf{P}$ on $\sum_{k=0}^{K} \mathbf{M}_k + \mathbf{B} + \mathbf{D} + \mathbf{Mask}$ . Formally,

$$\mathbf{P} = \text{Gumbel-Sinkhorn}\left(\sum_{k=0}^{K} \mathbf{M}_k + \mathbf{B} + \mathbf{D} + \mathbf{Mask}\right). \tag{6}$$

By multiplying the resulting soft permutation matrix $\mathbf{P}$ element-wise with the argument of Eq. 6, we obtain our approximation of the GED, as

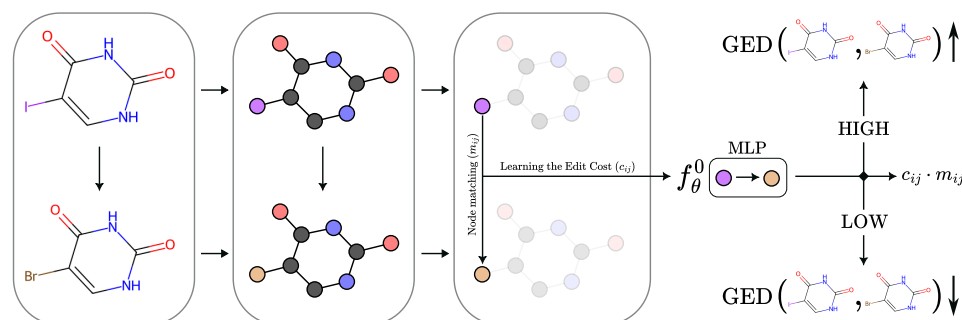

Figure 2: Illustrative example of cost learning via $f_\theta^{(k)}$. Here, a single-node difference drives the dissimilarity, and the substitution cost is learned to align the GED with the functional discrepancy.

$$\text{GED}(G, G') \approx \mathbf{P} \odot \left( \sum_{k=0}^{K} \mathbf{M}_k + \mathbf{B} + \mathbf{D} \right). \qquad (7)$$

We learn this GED approximation through an unsupervised optimization scheme, which refines the model's output without access to ground-truth GED values. The learning process is based on a key property of GED: it equals zero if, and only if, two graphs are identical (as stated in Eq. 4). This propriety provides an intrinsic signal that guides model optimization.

In our method, the representations generated by the GIN layers are projected onto a unit sphere, and the distance between node pairs representations is computed using cosine distance ranging from 0 to 1 (see Eq. 3). The optimization minimizes this distance for similar nodes, allowing the model to learn latent configurations consistent with the topological structure. Importantly, this process is governed by the geometry induced by spherical normalization, which constrains the evolution of the representations.

To further enhance learning effectiveness, we employ a pre-trained Gumbel-Sinkhorn method that preferentially selects node pairs representations with minimum distance. This directs optimization toward the most informative correspondences and encourages the model to capture relevant structural similarities. Overall, the approach behaves as a self-organizing system: the gradients derived from the GED minimization objective guide the adaptation of internal representations, promoting greater consistency between structurally similar graphs.

## 2.3 LEARNING THE EDIT COSTS

To enable learning of the edit costs, we define a series of $(K + 1)$ matrices $\mathbf{C}_k = (c_{ij})$, where each element $c_{ij}$ encodes the output of a learnable cost function $f_\theta^{(k)} : \tilde{V} \times \tilde{V}' \to \mathbb{R}_{>0}$ at level $k$. The parametric function $f_\theta^{(k)}$, implemented here as a Multi-Layer Perceptron (MLP) with a softplus activation ($\beta = 5$), produces strictly positive values $c_{ij}$. Using the soft permutation matrix $\mathbf{P}$ as in Eq. 6, the GED with learnable edit costs is then defined by

$$\text{GEDAN}(G, G') \approx \mathbf{P} \odot \left[ \lambda \left( \sum_{k=0}^{K} \mathbf{M}_k + \mathbf{B} + \mathbf{D} \right) + (1 - \lambda) \left( \sum_{k=0}^{K} (\mathbf{M}_k \odot \mathbf{C}_k \cdot \beta_k) \right) \right], \qquad (8)$$

where $\beta_k$ are learnable scalars controlling the contribution of each level $k$, and $\lambda$ is a scalar modulating the trade-off between respecting the topological structure and learning the edit costs.

Multiplying the values $c_{ij}$ with the corresponding matching value $m_{ij}$ ensures that in the case of a perfect match (i.e., $m_{ij} = 0$), the distance between representations remains unaltered by the learned cost (thereby preserving the consistency of the match itself – as illustrated in Figure 2).

Traditional GED approximation methods typically assign fixed scalar costs to edit operations, irrespective of node or edge type. Earlier approaches (e.g., Garcia-Hernandez et al. (2020)) attempt to

learn such costs only indirectly (e.g., via genetic algorithms), lacking direct optimization and differentiability. More advanced methods, such as GMSM (Pellizzoni et al., 2024), introduce node-specific costs but restrict them to the final matching step, without explicitly modeling granular edit differences. In contrast, our method learns fine-grained edit costs directly within a fully differentiable framework.

## 2.4 THE TRAINING OF GEDAN

GEDAN is designed to be a flexible framework and can therefore be trained in two distinct modes: unsupervised or supervised.

In the unsupervised setting, optimization follows Eq. 7 (see Appendix C.3 for the complete definition). If the weights of the functions $f_\theta^{(k)}$ are frozen and treated as fixed costs, GEDAN can be directly used to approximate GED, avoiding updates to the learned costs and preventing potential shortcut solutions. For consistency, the insertion and deletion costs ($a^\ominus, a^\oplus, b^\ominus, b^\oplus$) are also kept fixed.

In the supervised setting, where target labels are provided, the model leverages both the edit costs produced by the functions $f_\theta^{(k)}$ and the scalar parameters $a^\ominus, a^\oplus, b^\ominus, b^\oplus$ to align GED with the target labels. This gives GEDAN great flexibility. On the one hand, it can be used to minimize GED in a standard fashion using an appropriate loss (we use Smooth L1 loss (Girshick, 2015)). On the other hand, its most interesting application arises when topological dissimilarity does not coincide with functional dissimilarity. In this case, given the target labels, edit costs are optimized through a multi-task learning strategy (Song et al., 2016) that employs multiple loss functions. Multi-loss training in metric learning has been extensively studied (Wang et al., 2019a; Schroff et al., 2015) and is known to improve generalization and the robustness of learned representations. In our model, two complementary loss functions are employed.

The first is a *Softmax-based Contrastive Loss* (Wang et al., 2019b), which enables distance-based ranking without requiring an explicit margin. Given a query graph $G_Q$ as well as one positive key $G_P$ and $M$ negative keys $G_{N_1}, \ldots, G_{N_M}$ (i.e., a graph $G_P$ that is semantically similar to $G_Q$, and $M$ graphs that are semantically dissimilar to $G_Q$), the objective is to enforce that the GED between $G_Q$ and $G_P$ is lower than the GED between $G_Q$ and each of the $M$ negative keys $G_{N_1}, \ldots, G_{N_M}$. Formally,

$$\text{GED}(G_Q, G_P; \mathcal{C}) < \text{GED}(G_Q, G_{N_i}; \mathcal{C}) \quad \forall i \in [1, \ldots, M]. \tag{9}$$

This objective guides the model to learn edit costs $\mathcal{C}$ that align with semantic similarity, reducing the GED to positive keys while increasing it for negatives.

The second loss function is introduced to supervise the downstream task (Sobal et al., 2024), which may involve either regression or classification. This loss is computed via a neural network module that processes the predicted GEDs between the query graph and a set of key graphs, and uses this information to predict the label of the query graph. The complete definition of losses can be found in Appendix E.2.

## 3 EXPERIMENTAL EVALUATION

In this section we present the results obtained in three different experiments. The first evaluation aims at testing whether the novel model is able to approximate GED with sufficient accuracy in different cost scenarios. The second evaluation shows that GEDAN, by learning edit costs end-to-end, effectively aligns topological and functional distances. In contrast, methods that do not learn costs end-to-end become computationally impractical in these scenarios. Finally, in a third experiment, we show and discuss a use case of direct comparisons of molecular graphs using our model and the learned cost.

For the first experiment, we use three datasets from the TUDataset repository (Morris et al., 2020) that contain molecule graphs. The chosen datasets consist of intentionally small graphs, motivated by the need to ensure the feasibility of the supervised training required by most models. For the second and third evaluations, we employ two benchmark datasets (Wu et al., 2017; Dwivedi et al., 2023b), namely FreeSolv (Mobley & Guthrie, 2014) and BBBP (Martins et al., 2012), which describe hydration free energy and blood–brain barrier permeability, respectively.

Table 1: Average and standard deviation of the error and correlation of GED approximations to exact GED obtained with five different cost configurations. The best results per dataset and metric, grouped by training type (supervised (S) vs. unsupervised (U)), are highlighted in boldface, while the second-best results are underlined.

| | MODEL | AIDS | | | MUTAG | | | PTC MR | | |
|---|---|---|---|---|---|---|---|---|---|---|
| | | RMSE ($\downarrow$) | $\tau$ ($\uparrow$) | $\rho$ ($\uparrow$) | RMSE ($\downarrow$) | $\tau$ ($\uparrow$) | $\rho$ ($\uparrow$) | RMSE ($\downarrow$) | $\tau$ ($\uparrow$) | $\rho$ ($\uparrow$) |
| S | SimGNN [1] | $5.93 \pm 1.52$ | $\underline{0.556 \pm 0.03}$ | $0.765 \pm 0.04$ | $6.32 \pm 1.14$ | $0.597 \pm 0.06$ | $0.801 \pm 0.05$ | $5.38 \pm 1.47$ | $0.689 \pm 0.06$ | $0.855 \pm 0.05$ |
| | EGSC [47] | $4.63 \pm 1.40$ | $0.467 \pm 0.04$ | $0.643 \pm 0.04$ | $\mathbf{3.40 \pm 1.24}$ | $\mathbf{0.614 \pm 0.05}$ | $\mathbf{0.815 \pm 0.05}$ | $3.86 \pm 1.31$ | $\mathbf{0.726 \pm 0.04}$ | $\mathbf{0.893 \pm 0.03}$ |
| | GREED [48] | $\mathbf{3.05 \pm 0.08}$ | $0.429 \pm 0.08$ | $0.633 \pm 0.07$ | $4.87 \pm 0.48$ | $0.585 \pm 0.04$ | $0.643 \pm 0.06$ | $\mathbf{2.59 \pm 0.43}$ | $0.690 \pm 0.02$ | $\underline{0.859 \pm 0.02}$ |
| | ERIC [74] | $5.44 \pm 1.17$ | $0.475 \pm 0.05$ | $0.689 \pm 0.05$ | $6.05 \pm 1.54$ | $0.567 \pm 0.03$ | $0.787 \pm 0.04$ | $5.25 \pm 0.72$ | $0.688 \pm 0.05$ | $0.850 \pm 0.05$ |
| | GRAPHEDX$_{XOR}$ [29] | $4.69 \pm 0.69$ | $\mathbf{0.579 \pm 0.02}$ | $\mathbf{0.802 \pm 0.05}$ | $\underline{4.46 \pm 0.59}$ | $0.584 \pm 0.03$ | $0.789 \pm 0.06$ | $4.21 \pm 0.99$ | $0.691 \pm 0.04$ | $0.853 \pm 0.02$ |
| | GEDAN($\lambda = 0$) | $\underline{4.58 \pm 1.33}$ | $0.477 \pm 0.03$ | $0.696 \pm 0.03$ | $4.63 \pm 0.99$ | $0.579 \pm 0.03$ | $\mathbf{0.818 \pm 0.04}$ | $4.44 \pm 1.32$ | $\underline{0.693 \pm 0.05}$ | $0.856 \pm 0.04$ |
| U | APPROXIMATED GED [21] | $\underline{7.19 \pm 1.93}$ | $0.272 \pm 0.10$ | $0.402 \pm 0.15$ | $\underline{10.43 \pm 3.15}$ | $0.398 \pm 0.10$ | $0.556 \pm 0.13$ | $\underline{7.43 \pm 3.43}$ | $0.495 \pm 0.18$ | $0.661 \pm 0.24$ |
| | EUGENE [6] | $8.60 \pm 3.60$ | $\underline{0.433 \pm 0.10}$ | $\underline{0.581 \pm 0.15}$ | $11.63 \pm 6.20$ | $0.427 \pm 0.07$ | $0.608 \pm 0.07$ | $8.59 \pm 3.29$ | $0.593 \pm 0.12$ | $0.719 \pm 0.17$ |
| | GEDAN($\lambda = 1$) | $\mathbf{5.19 \pm 1.39}$ | $\mathbf{0.436 \pm 0.03}$ | $\mathbf{0.658 \pm 0.03}$ | $\mathbf{10.18 \pm 3.82}$ | $\mathbf{0.578 \pm 0.04}$ | $\mathbf{0.813 \pm 0.04}$ | $\mathbf{4.91 \pm 1.34}$ | $\mathbf{0.664 \pm 0.04}$ | $\mathbf{0.841 \pm 0.03}$ |

## 3.1 APPROXIMATION OF GRAPH EDIT DISTANCE

In the first experiment, we evaluate the quality of the GED approximations obtained by GEDAN, both supervised and unsupervised, and compare them with state-of-the-art models developed specifically for this task. The quality of the different GED approximations is evaluated by the *Root Mean Square Error* (RMSE) to the exact GED as well as Kendall's and Spearman's correlation values $\tau$ and $\rho$, respectively.

Table 1 shows the results obtained by the seven reference models as well as the two versions of our model. We report the mean and standard deviation of the results obtained with five different configurations[1], highlighting the best result per dataset and metric in boldface and underline the second best. In GEDAN, EUGENE, and GraphEdX, the costs of node insertion and deletion can be explicitly defined, while GEDAN and GraphEdX also support separate costs for edge insertion and deletion. These parameters are configured appropriately for each experimental setting.

Across all three datasets and evaluation metrics, GEDAN demonstrates competitive performance compared to state-of-the-art methods. In particular, the unsupervised version achieves the best results within its group, while the supervised variant – capable of using learned edit costs to approximate GED – performs comparably to existing approaches. We note, however, that supervised GEDAN is computationally more expensive than the other supervised methods and is primarily designed for scenarios where topological and functional similarity diverge. In the present benchmark, where the two measures coincide, it is not the optimal choice, yet it still delivers strong performance.

## 3.2 OPTIMIZATION OF EDIT COSTS

In this second evaluation, we study the effect of learning edit costs with GEDAN. While GEDAN can approximate GED, previous supervised models cannot be applied to this task because computing exact GED is infeasible and the optimal edit costs are not known a priori. Among theoretically feasible GED approaches, we consider grid search and genetic algorithms (Garcia-Hernandez et al., 2020). Both, however, face severe scalability issues when optimizing costs at the level of individual nodes or subgraphs, which limits their applicability. As a baseline, we therefore perform a grid search over 16 cost configurations (node and edge insertion/deletion costs set to 1 or 2) and report the best result. For unsupervised GED approximation, we adopt GEDAN, which offers flexibility by learning insertion and deletion costs separately, while providing a more practical trade-off between accuracy and efficiency than alternatives such as EUGENE (Bommakanti et al., 2024). For completeness, we also include both versions of GMSM (Pellizzoni et al., 2024), using comparable configurations to ensure a fair comparison.

We use the predicted GED to embed the graphs in a vector space using dissimilarity-based embeddings (Riesen & Bunke, 2009a). The prototype selection (actually required for graph embedding) is performed uniformly to ensure a fair comparison. Moreover, to ensure the robustness of our results and to avoid possible bias due to an unrepresentative selection of prototypes, we repeat the process three times (for each fold during cross-validation).

---

[1]These five different configurations represent five different combinations of fixed costs for deletions and insertions for both nodes and edges.

Table 2: Metrics computed on the predicted embeddings. The table reports mean values with standard deviations; best results are in boldface, and second-best results are underlined.

| | FreeSolv | | | | BBBP | | | |
| | $R^2(\uparrow)$ | | | | ROC-AUC $(\uparrow)$ | | | |
| | GEDAN Grid Search | GMSM [46] $(\epsilon \geq 0)$ | GMSM [46] $(\epsilon = 0)$ | GEDAN Learned | GEDAN Grid Search | GMSM [46] $(\epsilon \geq 0)$ | GMSM [46] $(\epsilon = 0)$ | GEDAN Learned |
|---|---|---|---|---|---|---|---|---|
| KNN | $0.247 \pm 0.12$ | $\mathbf{0.640 \pm 0.08}$ | $\underline{0.621 \pm 0.07}$ | $0.521 \pm 0.08$ | $0.634 \pm 0.11$ | $0.694 \pm 0.05$ | $\mathbf{0.733 \pm 0.06}$ | $\underline{0.696 \pm 0.05}$ |
| SVM | $0.281 \pm 0.14$ | $\underline{0.641 \pm 0.08}$ | $0.625 \pm 0.07$ | $\mathbf{0.682 \pm 0.08}$ | $0.656 \pm 0.09$ | $0.669 \pm 0.08$ | $\underline{0.668 \pm 0.09}$ | $\mathbf{0.705 \pm 0.07}$ |
| RF | $0.295 \pm 0.09$ | $\underline{0.603 \pm 0.06}$ | $0.601 \pm 0.09$ | $\mathbf{0.684 \pm 0.08}$ | $0.612 \pm 0.08$ | $0.692 \pm 0.07$ | $\mathbf{0.711 \pm 0.08}$ | $\underline{0.708 \pm 0.08}$ |
| MLP | $0.320 \pm 0.08$ | $\underline{0.668 \pm 0.08}$ | $0.656 \pm 0.11$ | $\mathbf{0.705 \pm 0.09}$ | $0.666 \pm 0.10$ | $0.728 \pm 0.09$ | $\underline{0.727 \pm 0.06}$ | $\mathbf{0.748 \pm 0.07}$ |
| Average | $0.286 \pm 0.03$ | $\underline{0.636 \pm 0.03}$ | $0.626 \pm 0.02$ | $\mathbf{0.648 \pm 0.09}$ | $0.642 \pm 0.02$ | $0.696 \pm 0.02$ | $\underline{0.710 \pm 0.03}$ | $\mathbf{0.714 \pm 0.02}$ |

Due to the large number of graphs, we select only 16 prototypes from the training set for each dataset. As a result, we obtain 16-dimensional embeddings for both the training and test sets, which are then used to solve the underlying regression or classification task of each dataset using standard models. Specifically, we employ $k$-Nearest Neighbors (KNN), Support Vector Machines (SVM), Random Forests (RF), and Multi-Layer Perceptrons (MLP), evaluating their performance using $R^2$, ROC-AUC, which are the metrics used in prior work (Xu et al., 2018).

Table 2 shows that learning edit costs enables GEDAN to produce GED approximations that translate into stronger downstream performance. Across multiple models, GEDAN achieves higher $R^2$ on FreeSolv and higher ROC-AUC on BBBP, outperforming fixed-cost baseline.

These improvements underline the importance of learning cost structures rather than relying on fixed configurations. While more accurate GED approximations can improve graph matching and highlight structural differences, this alone does not guarantee better predictive performance without task-specific cost optimization. Fixed grid-search costs, though a reasonable baseline, are inherently limited by discrete choices and become computationally infeasible when extended to fine-grained node- or subgraph-level settings. By learning edit costs end-to-end, GEDAN – and similarly GMSM – aligns GED with task-relevant similarities, producing embeddings that capture semantic differences critical for downstream applications.

## 3.3 Analysis of Edit Costs

A key advantage of our method is the use of graph matching to compare graphs via edit costs. In the following analysis, higher costs are interpreted as higher relevance, and node importance is visualized using a color gradient: lighter tones indicate higher-cost (i.e., more relevant) transformations, while darker tones indicate lower-cost (i.e., less relevant) transformations.

There are several interpretation methods and inherently explainable architectures available. Among these, those most closely related to ours are GMSM (Pellizzoni et al., 2024) and GraphMaskExplainer (Schlichtkrull et al., 2021), which similarly provide explanations through cost analysis. In contrast, methods such as PGExplainer (Luo et al., 2020), GNNExplainer (Ying et al., 2019) and SubgraphX (Yuan et al., 2021) follow different interpretability paradigms, relying on perturbation or the identification of relevant subgraphs. Finally, there are architectures designed to be inherently interpretable, including GNAN (Bechler-Speicher et al., 2024) and GATv2 (Brody et al., 2022), which allow explanations to be obtained directly from the model outputs.

The three analyses in the following are done on the FreeSolv dataset, with the goal of estimating the hydration free energy (HFE). To ensure a fair comparison, we use a GNN with performance comparable to the state-of-the-art (Buterez et al., 2024), alongside GEDAN with the same architectural depth. The analysis aims to identify the molecular substructures that contribute most significantly to the target property, namely HFE.

In Figure 3 (a), we compare two specific molecules, labeled as $(i)$ and $(ii)$. This type of comparison is only possible by approaches that explicitly learn edit costs, as they enable direct graph-to-graph matching. We present the result of both GEDAN and GMSM with respect to the HFE target. The two molecules are structurally similar; in particular, substituting an iodine (I) atom with a bromine (Br) atom results in only a minor change in HFE — 0.55 kcal/mol according to the reference values. Both GEDAN and GMSM capture this variation, with their edit-cost analyses correctly localizing the structural change responsible for the observed energy difference. Our model is fine-grained and

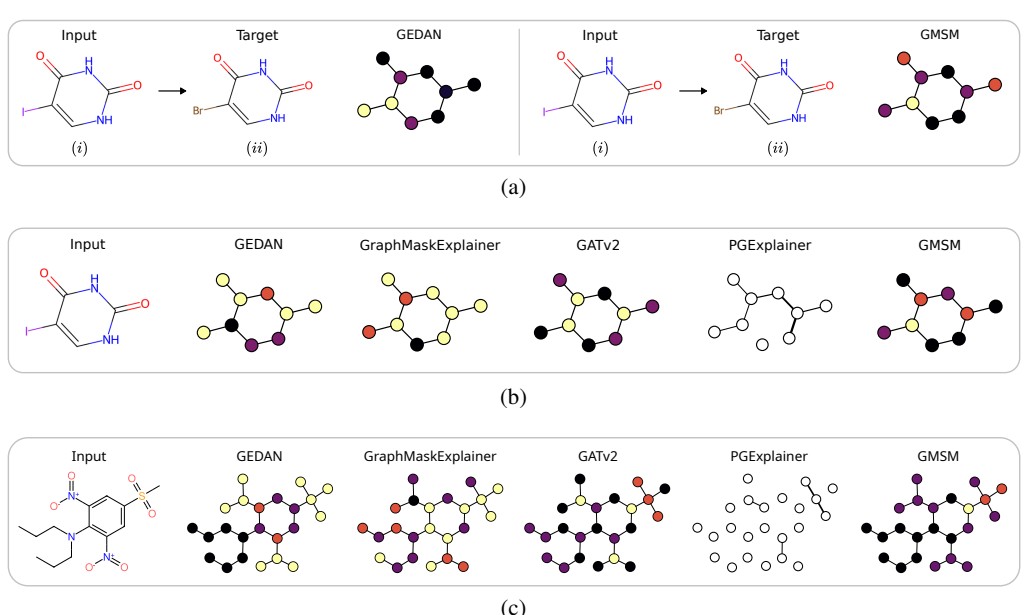

Figure 3: Cost analysis of models on the FreeSolv dataset. (a) shows a direct comparison between two molecules using GEDAN, while (b) and (c) show the overall cost analyses.

additive formulation provides a more precise costs modulation, which in turn facilitates a localized identification of structural differences. In contrast, GMSM also detects the modification but, due to its architectural design, provides a more global perspective on the change while remaining consistent with the observed variation.

In Figure 3 (b), we visualize the transformation costs between the molecule $(i)$ with respect to all graphs from the dataset. Most models correctly identify the two carbonyls as important for the target property. However, only our model, as well as the two reference models GMSM and GATv2, emphasize the chemically relevant symmetry of the functional group formed by the two carbonyls. In our model, the symmetry arises because the edit costs depend on the correspondences of the nodes in the additive layers, which promotes consistency of predictions when the structures are similar.

In Figure 3 (c), we examine a molecule containing functional groups associated with enhanced water solubility. In particular, the sulfonyl ($-SO_2-$) and nitro ($-NO_2$) groups, which have been reported to influence solubility (Goldberg & Parker, 1985; Morrison & Boyd, 1992), are correctly highlighted by all models, albeit with different levels of resolution and local precision.

## 4  CONCLUSION

Optimizing edit costs to align GED with functional dissimilarity between graphs is a computationally complex problem, exacerbating the already NP-hard nature of GED.

In this work, we propose a novel graph-based architecture that learns edit costs end-to-end, directly aligning GED with downstream tasks and overcoming the limitations of fixed-cost approaches. Although unsupervised GED approximation can improve graph matching in principle, achieving meaningful alignment with task-specific functional dissimilarity requires learning the edit costs. Our method enables fine-grained cost optimization at the node and subgraph level, producing embeddings that capture task-relevant differences and highlighting functionally important regions within the graphs. This fine-grained alignment not only improves predictive performance but also supports interpretable analysis by revealing which graph regions contribute most to the task.

Future work will focus on scaling the approach to larger graphs, extending cost learning to other graph types, and further investigating the theoretical properties of learned costs, aiming to provide bounds or monotonicity guarantees.

**Limitations** Our method has at least two major limitations. First, GEDAN introduces a greater computational overhead than recent approaches, due to the explicit management of cost and matching matrices, and the padding required to handle graphs of variable sizes. Despite optimizations (e.g., broadcasting), this results in increased execution time and memory requirements. Second, the model is currently limited to graphs with a maximum of 128 nodes, due to computational constraints associated with the use of the Gumbel-Sinkhorn network. Although this network is theoretically applicable to larger graphs, efficiency can degrade rapidly as size increases.

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

## A    IDENTITY CONDITION

As stated in Section 2, let $G = (\tilde{V}, E, \mu)$ and $G' = (\tilde{V}', E', \mu')$ be two graphs enriched with dummy nodes, we define the $k$-th level representation as in Eq. 2

$$h^{(k)}(v) = \begin{cases} \mu(v), & \text{if } k = 0, \\ A^k \left( h^{(k-1)}(v) \right) + h^{(k-1)}(v), & \text{if } k > 0. \end{cases}$$

Next, in Eq. 3, we define the distance $d^{(k)} : \tilde{V} \times \tilde{V}' \to (0, 1)$ as

$$d^{(k)}(v, v') = \frac{1}{2} \left( 1 - \frac{\langle h^{(k)}(v), h^{(k)}(v') \rangle}{\| h^{(k)}(v) \| \cdot \| h^{(k)}(v') \|} \right)$$

and in Eq. 4 the multi-scale distance

$$d^K(v, v') = \sum_{k=0}^{K} d^{(k)}(v, v').$$

The function $d^K(v, v')$ satisfies the following identity property:

$$d^K(v, v') = 0 \iff sub^{(K)}(G, v) \cong sub^{(K)}(G', v'), \tag{10}$$

where $sub^{(K)}(G, v)$ is the $K$-hop neighborhood graph anchored at node $v \in \tilde{V}$ which contains all nodes that have shortest path length at most $K$ to node $v$, and $\cong$ indicates that there exists a label-preserving graph isomorphism.

PROOF SKETCH

- **Case 1:** If $\mu(v) \neq \mu'(v')$, then $h^{(0)}(v) \neq h^{(0)}(v')$, and hence $d^{(0)}(v, v') > 0$, which implies $d^K(v, v') > 0$.
- **Case 2:** If $\mu(v) = \mu'(v')$ but $sub^{(K)}(G, v) \ncong sub^{(K)}(G', v')$, the injectivity of $A^k$ ensures that structural differences will be reflected in the representations at some $k \leq K$. To prevent over-smoothing, where embeddings become indistinguishable as $k$ increases, we use residual connections in Eq. 2. As demonstrated in Scholkemper et al. (2024), this strategy preserves discriminative power across layers. Therefore, there will exist at least one $k$ such that $h^{(k)}(v) \neq h^{(k)}(v')$, implying $d^K(v, v') > 0$.

## B    FORMAL STRUCTURES OF THE MATRICES

In the body of the paper, we stated that GEDAN uses five distinct matrices to learn edit costs. Figure 1 (a) shows a schematic representation of their structure. In this section, we provide a formal and more detailed description of these matrices.

$\mathbf{D}_{[n+m] \times [m+n]}$, captures the insertion $b^{\oplus}$ and deletion $b^{\ominus}$ costs for edges, computed as:

$$dg(v_i, v_j') = \text{ReLU}(\deg(v_i) - \deg(v_j')) \cdot b^{\ominus} + \text{ReLU}(\deg(v_j') - \deg(v_i)) \cdot b^{\oplus}. \tag{11}$$

The corresponding matrix $\mathbf{D}$ is formally defined as:

$$\mathbf{D} = \left[ \begin{array}{ccc|cc} dg(v_0, v_0') & \cdots & dg(v_0, v_m') & dg(v_0, \epsilon) & \infty \\ \vdots & \ddots & \vdots & & \ddots \\ dg(v_n, v_0') & \cdots & dg(v_n, v_m') & \infty & dg(v_n, \epsilon) \\ \hline dg(\epsilon, v_0') & & \infty & & \\ & \ddots & & & 0 \\ \infty & & dg(\epsilon, v_m') & & \end{array} \right]. \tag{12}$$

$\mathbf{B} = (b_{ij})_{[n+m] \times [m+n]}$, encodes the insertion $a^{\oplus}$ and deletion $a^{\ominus}$ costs for nodes

$$b_{ij} = \begin{cases} a^{\ominus}, & \text{if } i < n \text{ and } j \geq m, \\ a^{\oplus}, & \text{if } i > n \text{ and } j \leq m, \\ 0 & \text{else}, \end{cases} \tag{13}$$

it is defined as:

$$\mathbf{B} = \left[ \begin{array}{cc|cc} & & a^{\oplus} & \infty \\ & 0 & & \ddots \\ & & \infty & a^{\oplus} \\ \hline a^{\ominus} & \infty & & \\ \ddots & & & 0 \\ \infty & a^{\ominus} & & \end{array} \right]. \tag{14}$$

$\mathbf{M}_{k[n+m] \times [m+n]}$ denotes the matrix of pairwise node dissimilarities based on their $k$-hop representations. Formally, we use the following structure

$$\mathbf{M}_k = \left[ \begin{array}{ccc|cc} d^{(k)}(v_0, v_0') & \cdots & d^{(k)}(v_0, v_m') & d^{(k)}(v_0, \epsilon) & \infty \\ \vdots & \ddots & \vdots & & \ddots \\ d^{(k)}(v_n, v_0') & \cdots & d^{(k)}(v_n, v_m') & \infty & d^{(k)}(v_n, \epsilon) \\ \hline d^{(k)}(\epsilon, v_0') & & \infty & & \\ & \ddots & & & 0 \\ \infty & & d^{(k)}(\epsilon, v_m') & & \end{array} \right]. \tag{15}$$

$\mathbf{C}_{k[n+m] \times [m+n]}$ contains learnable transformation costs between node embeddings at level $k$, formally,

$$\mathbf{C}_k = \left[ \begin{array}{ccc|cc} f^{(k)}(v_0, v_0') & \cdots & f^{(k)}(v_0, v_m') & f^{(k)}(v_0, \epsilon) & \infty \\ \vdots & \ddots & \vdots & & \ddots \\ f^{(k)}(v_n, v_0') & \cdots & f^{(k)}(v_n, v_m') & \infty & f^{(k)}(v_n, \epsilon) \\ \hline f^{(k)}(\epsilon, v_0') & & \infty & & \\ & \ddots & & & 0 \\ \infty & & f^{(k)}(\epsilon, v_m') & & \end{array} \right]. \tag{16}$$

We could extend the formulation into a temperature-scaled version by introducing a scalar $\mathcal{T}$, which is used to normalize the combined dissimilarity matrix $\sum_{k=0}^{K} \mathbf{M}_k$, and a weight normalization factor $\beta_K = \sum_{k=0}^{K} \beta_k$ applied to the cost aggregation $\sum_{k=0}^{K} \mathbf{C}_k$.

As defined in the main body, we use a pre-trained Gumbel-Sinkhorn network (Mena et al., 2018) to compute a soft permutation matrix $\mathbf{P}$. Incorporating temperature, this becomes:

$$\mathbf{P} = \text{Gumbel-Sinkhorn} \left[ \left( \sum_{k=0}^{K} \mathbf{M}_k \right) \cdot \frac{1}{\mathcal{T}} + \mathbf{B} + \mathbf{D} \right], \tag{17}$$

The unsupervised GED approximation, is defined as:

$$\text{GED}(G, G') \approx \mathbf{P} \odot \left[ \left( \sum_{k=0}^{K} \mathbf{M}_k \right) \cdot \frac{1}{\mathcal{T}} + \mathbf{B} + \mathbf{D} \right] + \lambda_{\mathcal{T}} \cdot log(\mathcal{T}). \tag{18}$$

while GEDAN becomes

$$\text{GEDAN}(G, G') = \mathbf{P} \odot \left[ \lambda \left( (\sum_{k=0}^{K} \mathbf{M}_k) \cdot \frac{1}{\mathcal{T}} + \mathbf{B} + \mathbf{D} \right) + (1 - \lambda) \left( (\sum_{k=0}^{K} (\mathbf{M}_k \odot \mathbf{C}_k \cdot \beta_k)) \cdot \frac{1}{\beta_K} \right) \right] + \lambda_{\mathcal{T}} \cdot log(\mathcal{T}), \tag{19}$$

where $\lambda_{\mathcal{T}}$ modulates the effect of the logarithm of the temperature $\mathcal{T}$. The temperature regulates the "softness" of the matching matrices: higher values introduce greater uncertainty into the optimal matching, favoring exploration during optimization; conversely, lower values make the matrices more rigid, accelerating convergence but increasing the risk of converging towards local minima and suboptimal matches. The parameter $\lambda_{\mathcal{T}}$ helps stabilize optimization as a function of temperature.

## C  Model Details

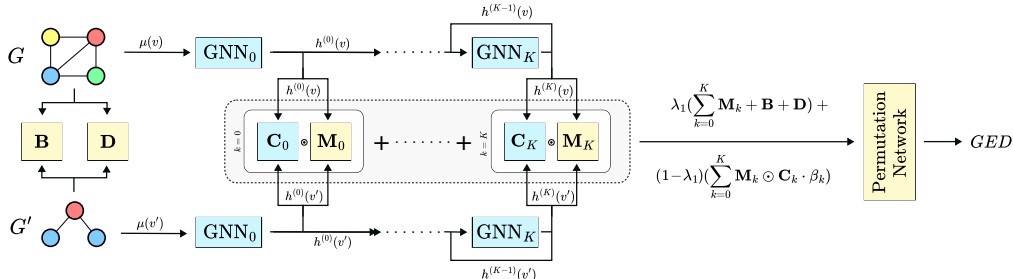

Figure 4: The architectural scheme of GEDAN

To implement the proposed method, some preprocessing operations need to be performed on the data. The method requires bipartite graphs, which results in the use of square matrices. To fulfil this constraint, additional nodes with dummy representation are added to each graph.

In addition to these structural nodes, additional padding nodes are introduced to ensure that the matrices not only remain square, but also reach a pre-defined size (e.g. $64 \times 64$ or $128 \times 128$). These padding nodes also have a dummy representation, used exclusively to adjust the size of the matrix.

To manage this structure, binary masks that selectively obscure certain pairs of nodes are computed and applied. However, this strategy is not an optimal solution, both because of the increase in memory space required and the introduction of computationally unnecessary computations. Improvements in this regard are in future work.

GIN layers are implemented using a simple MLP, followed by a ReLU activation function to introduce non-linearity. The output of each layer is then normalized using LayerNorm. Before calculating distances, the embedding vectors are normalized, resulting in representations on a unit sphere (spherical embeddings).

As for the functions $f^{(k)} \rightarrow \mathbf{C}_k$, we concatenate the representations of the two nodes involved and provide the result as input to an MLP. This MLP employs a ReLU as an intermediate activation function and applies a softplus function with parameter $\beta = 5$ on the final output, as described in the main body of the text.

We use the Adam optimizer. The learning rate, number of layers $K$, embedding size, and batch size are chosen according to the dataset and type of model considered. We explored several configurations, and in the main paper we use those that offer the best trade-off between accuracy and computational cost.

Training GEDAN requires substantial computational resources. We employ a NVIDIA A100 GPUs for GED approximation, while experiments on learning edit costs are conducted on a cluster with two NVIDIA H100 GPUs to optimize training time and cost. The implementation is based on PyTorch (v2.2.0) together with the PyTorch Geometric library (v2.4.0), with hardware acceleration provided by CUDA (v12.1). The source code is publicly available at https://github.com/gedan-iclr2026/GEDAN.

### C.1  Illustrative Graph Matching Example with $M_k$ Penalties

In Fig. 1 we illustrate how our model evaluates the matching between two graphs. To provide a concrete example, we consider a real case from the PTC MR dataset (Morris et al., 2020), where two

structurally similar graphs are compared. This example highlights how the additive penalty improves the quality of the matching.

Fig. 5 reports this real-world instance. For clarity of visualization, we restrict the matching matrix to the first eight nodes. In this setting, nodes 0 and 5 of graph $G$ can potentially match with nodes 1 and 4 of graph $G'$. Importantly, we stress that the analysis here is based solely on the matrices $\mathbf{M}_k$, without directly considering node degrees.

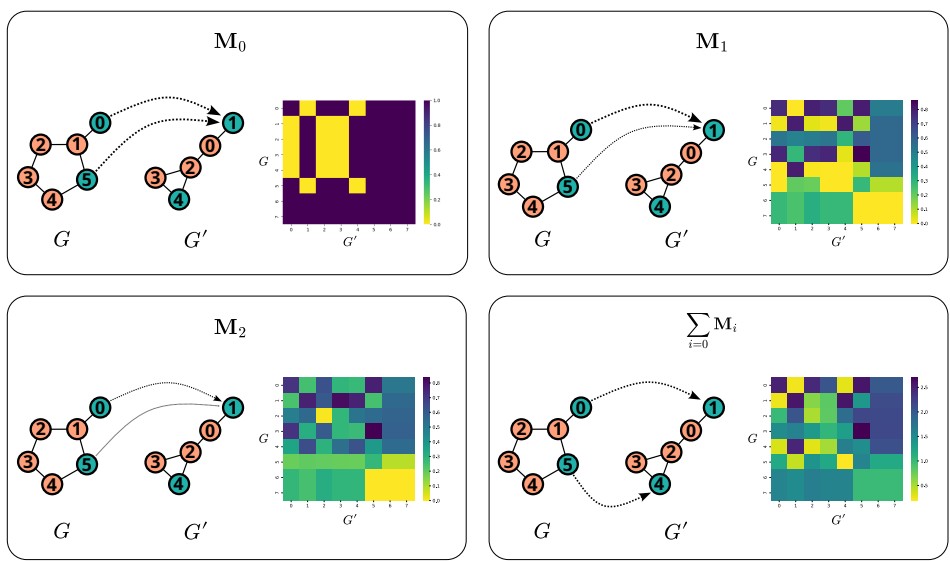

Figure 5: Matching example on PTC MR using $\mathbf{M}_k$ matrices.

At layer $\mathbf{M}_0$, which relies exclusively on node labels, we observe that nodes 0 and 5 of $G$ perfectly match with nodes 1 and 4 of $G'$, respectively. Moving to $\mathbf{M}_1$, where neighborhood information is incorporated, clear differences emerge. Specifically, node 0 in $G$ has exactly one pink neighbor, while node 5 has two. A similar configuration is present in $G'$, with node 1 having one pink neighbor and node 4 having two. Consequently, in the matching matrix, entry $\mathbf{M}_{1_{0,1}}$ exhibits a bright yellow color (distance 0), while $\mathbf{M}_{1_{0,4}}$ appears green (higher distance). Conversely, node 5 in $G$ yields a bright yellow value at $\mathbf{M}_{1_{5,4}}$, reflecting a nearly identical representation.

Proceeding to $\mathbf{M}_2$, additional refinements occur: similarity scores change for most nodes, while only the pair of nodes labeled 2 in both graphs retain a strong correspondence. Finally, when aggregating $\mathbf{M}_T = \sum_{i=0} \mathbf{M}_i$, the result clearly emphasizes the bright yellow values at $\mathbf{M}_{T_{0,1}}$ and $\mathbf{M}_{T_{5,4}}$, thereby reinforcing the correct matchings.

## C.2 IMPACT OF THE TEMPERATURE $\mathcal{T}$

As shown in Formula 18, the approximation can be extended with a temperature parameter $\mathcal{T}$, which regulates the learning of the optimal matching. This is particularly useful since the parameter $K$ cannot always cover the maximum depth of the graphs. Allowing the model an initial exploration by softening the distance matrices enables the selection of non-ideal node pairs at early stages. Although this may appear counterintuitive, GIN networks update their representations differently under such conditions, and non-optimal initial matchings can lead to better solutions, precisely because the full graph structure is not observable.

To illustrate this effect, we consider graphs from the ENZYMES dataset (Morris et al., 2020) with up to 64 nodes. We employ a $128 \times 128$ Gumbel-Sinkhorn module and restrict the number of layers to $k = 3$. In this setting, a significant portion of the graph structure remains unobserved. Different values of $\mathcal{T}$ soften the matching matrix to varying degrees, thus promoting different exploration strategies and optimization trajectories. Performance is evaluated against the lower bound of the GED, defined as the minimum number of operations that must occur when transforming one graph into another, i.e., the absolute difference in the number of nodes and edges.

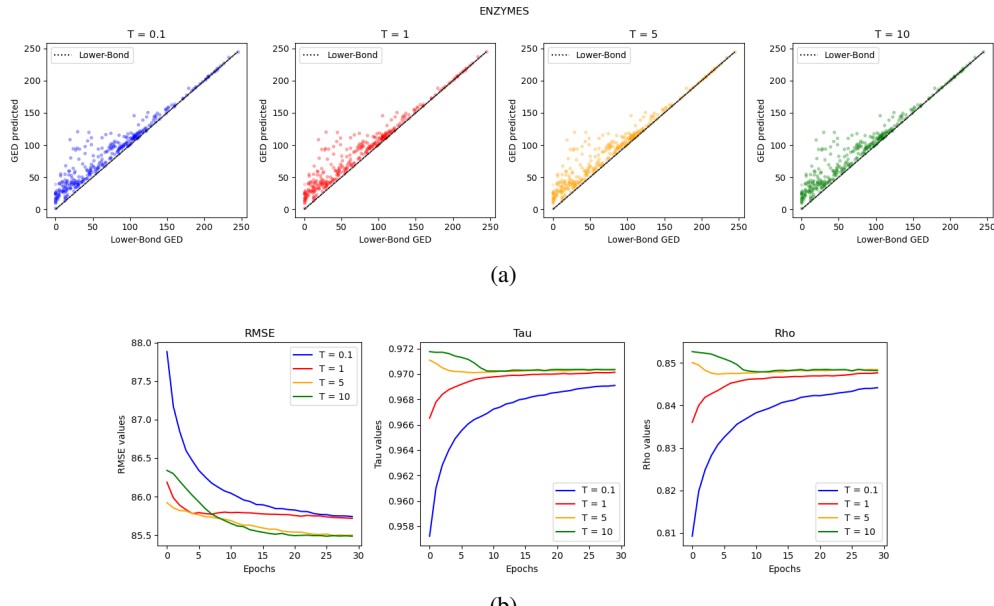

Figure 6: Effect of the temperature $\mathcal{T}$ on graph matching, showing how softening the distance matrix enables different exploration strategies and optimization outcomes.

### C.3 THE LOSS TO APPROXIMATE THE GRAPH EDIT DISTANCE

As described in the main text, GEDAN($\lambda = 1$) is trained in an unsupervised manner to approximate the *Graph Edit Distance* (GED) by optimizing its internal node representations without relying on ground-truth distances. The associated loss function is defined as

$$\omega^* = \arg\min_{\omega} \sum f_{\omega}(G, G'), \tag{20}$$

where $\omega$ denotes the parameters of the GINs generating node representations $h^{(k)}(v)$ and $h^{(k)}(v')$ for nodes $v \in G$ and $v' \in G'$, respectively. GEDAN ($f_{\omega}$) estimates the GED by leveraging the fact that it is zero if and only if two graphs are identical (see Eq. 10). This property provides an intrinsic learning signal, enabling unsupervised training.

This signal underpins the stability of learning: for two graphs $G$ and $G'$ that are identical up to the $k$-WL test, there exists a node-wise matching such that corresponding representations at layer $k$ are identical, i.e., their pairwise distances are zero. The Gumbel-Sinkhorn module naturally selects these pairs as having minimal distance, preventing the model from altering them.

The multi-scale mechanism further reinforces meaningful alignment by penalizing differences between node representations: distances vary between 0 (identical) and 1 (maximally different). Nodes not belonging to the same graph incur higher assignment costs, while nodes that remain similar across scales up to layer $k$ are assigned lower costs and thus preferentially matched. This effectively encourages alignment of embeddings that are intrinsically similar, as measured by multi-scale distances.

To ensure stability and avoid trivial solutions, we impose two constraints on the learned representations. First, we use spherical embeddings to prevent the model from minimizing pairwise distances simply by shrinking norms. Second, we pre-train the assignment matrix $\mathbf{P}$, since training it from scratch can introduce gradient noise or allow trivial shortcuts. Pre-training ensures that the network has already learned a discrete matching strategy; freezing $\mathbf{P}$ during subsequent training forces the network to minimize the loss solely through the embeddings. Moreover, to prevent other shortcuts, we do not optimize the Gumbel-Sinkhorn network nor the deletion/insertion costs $a^{\ominus}, a^{\oplus}, b^{\ominus}, b^{\oplus}$, effectively setting $\lambda = 1$ and excluding the use of matrices $\mathbf{C}_k$. Without these constraints, the

network could trivially minimize the loss by adjusting these components, bypassing meaningful representation learning.

In summary, this framework encourages alignment of similar nodes by minimizing the total assignment cost through their embeddings. The matching mechanism itself remains fixed; embeddings "self-organize" under gradient descent to reduce the total cost. The loss attains a guaranteed minimum of zero if and only if the two graphs are identical, or equivalently, indistinguishable under the *k-Weisfeiler-Lehman* test.

## C.4    GUMBEL-SINKHORN

To obtain a good $\mathbf{P}$, we modify the version of the original Gumbel-Sinkhorn network (Mena et al., 2018; Shah et al., 2024), using the Truncated Gumbel (Neamah & Qasim, 2021) to reduce the gradient variance and add stability. We then sample with the Gumbel-Max Trick (Huijben et al., 2023) while maintaining differentiability. Finally we stabilize the normalization of Sinkhorn (Sinkhorn, 1964) with an early stop, improving efficiency and limiting gradient vanishing (LeCun et al., 1998).

The Gumbel-Sinkhorn network has been trained for 200 epochs on matrices of sizes $32 \times 32$, $64 \times 64$, $128 \times 128$ and $256 \times 256$. The matrices used for training are randomly generated, with 250,000 samples for each of the three dimensions. For each matrix, *Linear Sum Assignment* (LSA) is calculated and used as a reference label. Optimization is performed by minimizing the Huber Loss, with a learning rate of 0.001 and using the Adam optimizer. Training also includes a curriculum learning phase, during which many matrices with zero values on the diagonal are initially proposed, a strategy that facilitated model convergence. In addition, an early stopping mechanism is applied on the Sinkhorn process, with a limit of 50 iterations, where the stopping condition is determined by a difference between consecutive iterations of less than 1e-4.

To test the performance of the Gumbel-Sinkhorn network, 10000 new random matrices are generated, for which the LSA varies between 0.001 and 199.750, with an average of 77.375. The overall performance of the model calculated on the $64 \times 64$ matrices using high temperatures has a $\tau$ value of 0.991, a $R^2$ coefficient of 0.993 and a Root Mean Square Error (RMSE) of 3.940. These configurations are kept stable in the pre-trained versions in GEDAN.

To assess the importance of pre-training (PT), we reproduce the results of Table 10 using a Gumbel-Sinkhorn module without pre-training (No-PT). In this setting, the network bears a heavier optimization load: although the model attempts to minimize the matching while jointly training the module, the dual optimization produces noisier and less accurate gradients. This effect is particularly critical in the unsupervised setting, where the model must self-organize; noisy gradients can destabilize the process and lead to highly suboptimal solutions. In contrast, under supervised training the impact is less severe, as shown in GraphEdX$_{XOR}$ (Jain et al., 2024), where the module is not pre-trained but remains viable thanks to the stable guidance provided by labels.

Table 3: Ablation study on the effect of pre-training (PT) the Gumbel-Sinkhorn module versus no pre-training (No-PT) on Configuration 1. Best results for each metric and dataset are highlighted in boldface.

| MODELS | | AIDS [39] | | | MUTAG [39] | | | PTC MR [39] | | |
|---|---|---|---|---|---|---|---|---|---|---|
| | | RMSE ($\downarrow$) | $\tau$ ($\uparrow$) | $\rho$ ($\uparrow$) | RMSE ($\downarrow$) | $\tau$ ($\uparrow$) | $\rho$ ($\uparrow$) | RMSE ($\downarrow$) | $\tau$ ($\uparrow$) | $\rho$ ($\uparrow$) |
| PT | GEDAN($\lambda = 0$) | $\mathbf{2.82 \pm 0.19}$ | $\mathbf{0.484 \pm 0.05}$ | $\mathbf{0.694 \pm 0.05}$ | $\mathbf{5.20 \pm 0.15}$ | $\mathbf{0.650 \pm 0.04}$ | $\mathbf{0.887 \pm 0.01}$ | $\mathbf{3.18 \pm 0.65}$ | $\mathbf{0.655 \pm 0.08}$ | $\mathbf{0.826 \pm 0.07}$ |
| | GEDAN($\lambda = 1$) | $3.54 \pm 0.14$ | $0.437 \pm 0.06$ | $0.652 \pm 0.06$ | $13.30 \pm 0.85$ | $0.611 \pm 0.09$ | $0.852 \pm 0.04$ | $3.51 \pm 0.31$ | $0.649 \pm 0.07$ | $0.823 \pm 0.05$ |
| No-PT | GEDAN($\lambda = 0$) | $6.973 \pm 3.89$ | $0.193 \pm 0.10$ | $0.254 \pm 0.14$ | $916.82 \pm 8.83$ | $0.603 \pm 0.05$ | $0.723 \pm 0.05$ | $5.52 \pm 1.75$ | $0.445 \pm 0.09$ | $0.591 \pm 0.08$ |
| | GEDAN($\lambda = 1$) | $2509.84 \pm 18.5$ | $0.074 \pm 0.04$ | $0.101 \pm 0.07$ | $3224.39 \pm 33.8$ | $0.596 \pm 0.05$ | $0.722 \pm 0.05$ | $2335.98 \pm 121.9$ | $0.396 \pm 0.07$ | $0.584 \pm 0.08$ |

The computational complexity of the Gumbel–Sinkhorn module is $O(Ln^2)$, where $L$ denotes the number of Sinkhorn normalization iterations. This is lower than the complexity of the Kuhn–Munkres algorithm ($O(n^3)$) (Kuhn, 1955). In the case of our model (GEDAN), choosing the number of iterations such that $L \ll n$ keeps the computational cost manageable even for large matrices. Recent works, such as Tang et al. (2024), further demonstrate that this complexity can be reduced to $\approx O(n^2)$, providing strategies that are particularly useful for improving the efficiency of our approach.

## C.5 INFLUENCE OF DEPTH AND INFERENCE DYNAMICS

In this section, we demonstrate how increasing the number of layers and optimizing their structure can lead to improved GED prediction. As suggested by Eq. 10, the hyperparameter $K$ plays a crucial role by controlling the receptive field size over the graph. A higher value of $K$ allows the model to capture more structural information, which should, in principle, lead to better graph-aware predictions.

To isolate the effect of architectural depth, we conduct a first experiment using GEDAN($\lambda = 1$) in a purely inference setting, without any prior training. As shown in Table 4, increasing $K$ leads to both a higher parameter and consistent improvements in rank-based metrics $\tau$ and $\rho$. This suggests enhanced alignment with the underlying graph structure. However, this increase in structural coherence does not directly translate to better RMSE performance. We hypothesize that this is due to the nature of the injective $A^k$ layers, which are implemented as non-linear GIN layers. While such $K$ layers improve relational consistency, they do not necessarily minimize prediction error in an untrained setting.

Table 4: GEDAN($\lambda = 1$) inference of the GED approximation with $K$-layers on Configuration 1.

| LAYERS | | AIDS [39] | | | MUTAG [39] | | | PTC MR [39] | | |
|---|---|---|---|---|---|---|---|---|---|---|
| K | AVG. PARAM. | RMSE ($\downarrow$) | $\tau$ ($\uparrow$) | $\rho$ ($\uparrow$) | RMSE ($\downarrow$) | $\tau$ ($\uparrow$) | $\rho$ ($\uparrow$) | RMSE ($\downarrow$) | $\tau$ ($\uparrow$) | $\rho$ ($\uparrow$) |
| 1 | 6K | **6.71 $\pm$ 0.07** | 0.384 $\pm$ 0.07 | 0.600 $\pm$ 0.07 | 11.52 $\pm$ 0.74 | 0.602 $\pm$ 0.09 | 0.844 $\pm$ 0.05 | **4.92 $\pm$ 0.52** | 0.585 $\pm$ 0.08 | 0.785 $\pm$ 0.06 |
| 2 | 8K | 6.79 $\pm$ 0.13 | 0.393 $\pm$ 0.07 | 0.609 $\pm$ 0.07 | 11.77 $\pm$ 0.80 | 0.605 $\pm$ 0.09 | 0.848 $\pm$ 0.04 | 4.96 $\pm$ 0.52 | 0.598 $\pm$ 0.08 | 0.795 $\pm$ 0.06 |
| 4 | 12K | 6.80 $\pm$ 0.11 | 0.404 $\pm$ 0.07 | 0.609 $\pm$ 0.07 | 11.40 $\pm$ 0.93 | 0.612 $\pm$ 0.09 | 0.855 $\pm$ 0.04 | 5.26 $\pm$ 0.57 | 0.605 $\pm$ 0.08 | 0.801 $\pm$ 0.06 |
| 8 | 19K | 7.20 $\pm$ 0.23 | 0.412 $\pm$ 0.07 | 0.630 $\pm$ 0.07 | 10.60 $\pm$ 0.89 | 0.618 $\pm$ 0.08 | 0.862 $\pm$ 0.04 | 5.70 $\pm$ 0.56 | 0.614 $\pm$ 0.07 | 0.809 $\pm$ 0.06 |
| 16 | 35K | 7.48 $\pm$ 0.14 | 0.422 $\pm$ 0.07 | 0.638 $\pm$ 0.07 | 10.15 $\pm$ 0.93 | 0.618 $\pm$ 0.08 | 0.868 $\pm$ 0.03 | 6.17 $\pm$ 0.60 | 0.621 $\pm$ 0.08 | 0.814 $\pm$ 0.06 |
| 32 | 66K | 7.91 $\pm$ 0.21 | **0.430 $\pm$ 0.07** | **0.647 $\pm$ 0.07** | 9.29 $\pm$ 1.12 | **0.615 $\pm$ 0.08** | **0.871 $\pm$ 0.03** | 6.64 $\pm$ 0.63 | **0.633 $\pm$ 0.08** | **0.823 $\pm$ 0.06** |

RMSE minimization is achieved through training, where the network adjusts its internal representations via gradient descent. In Fig. 7, we illustrate the training dynamics for $K = 8$ under Configuration 1 on the PTC MR dataset. In this case, training yields an RMSE of $4.087 \pm 0.395$, with $\tau = 0.624 \pm 0.071$ and $\rho = 0.808 \pm 0.057$. Notably, this corresponds to an RMSE improvement of approximately 28.30%.

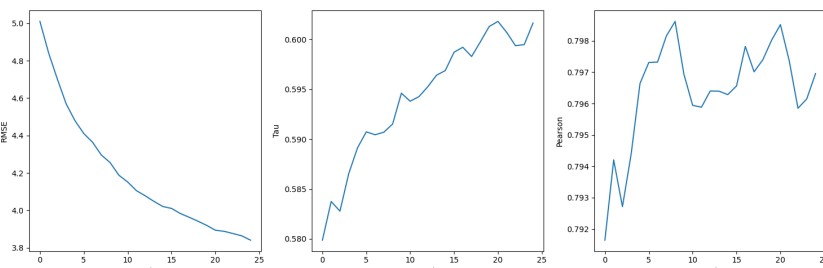

Figure 7: The training curves of GEDAN($\lambda = 1$) on the test-set of RMSE, $\tau$ and $\rho$ respectively.

An additional analysis can be performed by observing the scatter plot between the ground-truth GED and the predicted values across training epochs. Over the 25 training epochs, we observe the evolution in the model's internal representations, which directly affects its predictions.

In Fig. 8, we report the scatter plots at three stages of training: (a) epoch 1, (b) epoch 13, and (c) epoch 25. Each point represents a graph pair. The red arrow highlights a specific graph pair whose prediction significantly improves over time, moving closer to the identity line.

The illustrative case, highlighted by the red arrow, is a graph pair with a very low ground-truth GED, initially predicted with a RMSE greater than 10. As training progresses, the predicted GED for this graph pair gradually decreases, moving closer to the identity line. This behavior demonstrates the model's ability to refine its internal representation over time.

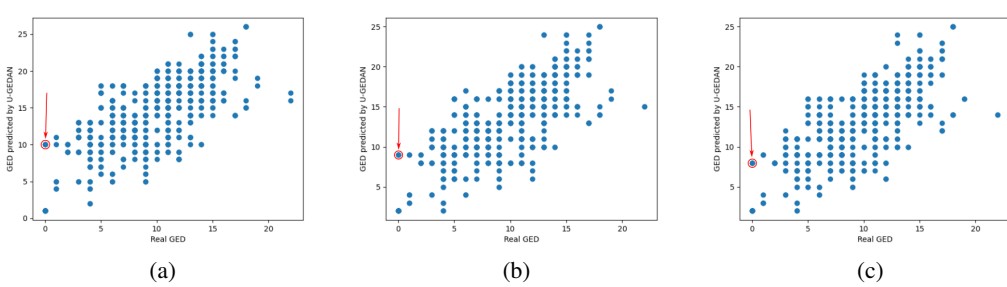

(a)  (b)  (c)

Figure 8: Scatter plots showing the GED predicted by GEDAN($\lambda = 1$) and ground-truth GED values at different training epochs: (a) epoch 1, (b) epoch 13, and (c) epoch 25.

This improvement is achieved in an unsupervised way, without direct supervision on the GED values, showing that GEDAN($\lambda = 1$) can internally structure representations that aligns with true graph distances. Further examples are shown in Figures 13, 14, 15 and 16.

## D DETAILS OF THE GED APPROXIMATION

In this section, we report the results obtained for each configuration used in our GED approximation experiments. We selected three benchmark datasets (Morris et al., 2020): AIDS, MUTAG, and PTC MR (see Table 5).

For computational feasibility, we restrict the graph sizes: only graphs with at most 9 nodes for AIDS, 16 nodes for MUTAG, and 10 nodes for PTC MR are included. This constraint is essential, as the precomputed labels used for training are based on GED values. Given that GED computation is NP-hard, these restrictions enable the generation of exact distances within a reasonable amount of time.

Table 5: Characteristics of the dataset used in the GED approximation.

| AIDS | | | MUTAG | | | PTC MR | | |
|---|---|---|---|---|---|---|---|---|
| N. GRAPH PAIRS | AVG. NODES | AVG. EDGES | N. GRAPH PAIRS | AVG. NODES | AVG. EDGES | N. GRAPH PAIRS | AVG. NODES | AVG. EDGES |
| 190,096 | 8.08 | 15.39 | 5,776 | 13.29 | 28.08 | 16,900 | 6.95 | 13.00 |

Furthermore, this setting allows us to build upon the recent analysis by Yang et al. (2024), which highlights the potential to achieve high-quality GED approximations even in low-label regimes. In particular, we use the MUTAG dataset to explore how models behave under data scarcity conditions. Specifically, the number of graph pairs used in our experiments is as follows: MUTAG contains 5,776 pairs, PTC MR contains 16,900 pairs, and AIDS contains 190,096 pairs.

### D.1 THE SETTING UP OF THE CONFIGURATIONS

We define five distinct cost configurations, summarized in Table 6, each modeling a different edit cost scenario for GED approximation.

1. The first configuration serves as a baseline, where all operations – node and edge insertions, deletions, and substitutions – have uniform costs set to 1, as commonly assumed in several models.

2. In the second configuration, the costs of inserting and deleting nodes are twice as expensive as the operations on edges.

3. In the third, edge insertions and deletions are assigned a cost twice as high as that of node operations.

4. In the fourth, insertions are twice as expensive as deletions.

5. In the fifth, deletions are twice as expensive as insertions.

Table 6: Operation costs of the datasets used in the GED approximation test.

| CONFIGURATIONS | TYPE | NODE INSERTION COST | NODE DELETION COST | EGDE INSERTION COST | EDGE DELETION COST |
|---|---|---|---|---|---|
| CONF. 1 | BASELINE | 1 | 1 | 1 | 1 |
| CONF. 2 | SYMMETRIC | 2 | 2 | 1 | 1 |
| CONF. 3 | SYMMETRIC | 1 | 1 | 2 | 2 |
| CONF. 4 | ASYMMETRIC | 2 | 1 | 2 | 1 |
| CONF. 5 | ASYMMETRIC | 1 | 2 | 1 | 2 |

For completeness, we report the statistics of GED values for each configuration and dataset considered. Recent work indicates that GED benchmarks may have some bias related to isomorphism, or that this bias may be easily introduced during the training phase (Roy et al., 2025).

In our case, since the main goal is to compare our model with those in the literature, we explicitly address the risk of structural leakage across datasets. To mitigate this issue, we verify that the GED is equal to zero only for pairs of identical graphs. Concretely, we inspect the full GED matrix and ensure that zero values appear exclusively along the diagonal. We then exclude from the analysis any graph that exhibits a zero GED with a different graph (i.e., extra-diagonal zeros). Figure 9 shows the resulting GED matrix for our three datasets.



(a) The GED value matrices in the AIDS dataset

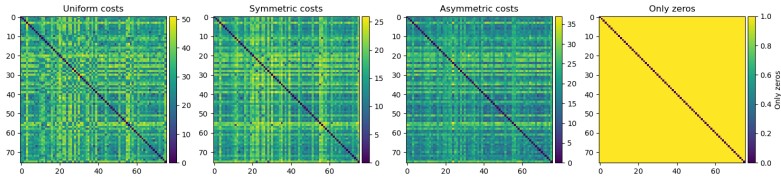

(b) The GED value matrices in the MUTAG dataset

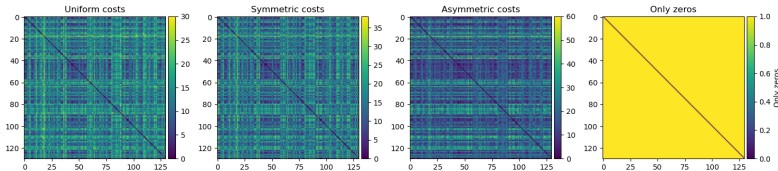

(c) The GED value matrices in the PTC MR dataset

Figure 9: The matrices of GED values for configurations 1, 2, and 4, showing where the GED values are equal to zero.

Table 7: Statistics of the AIDS dataset used in the GED approximation.

| DATASET | CONFIGURATIONS | MEAN | MEDIAN | STD | VARIANCE | MIN | MAX |
|---|---|---|---|---|---|---|---|
| AIDS (MORRIS ET AL., 2020) | 1 | 11.4 | 11.0 | 3.73 | 13.95 | 0.0 | 31.0 |
| | 2 | 13.3 | 13.0 | 4.83 | 23.33 | 0.0 | 37.0 |
| | 3 | 20.5 | 20.0 | 7.52 | 56.67 | 0.0 | 59.0 |
| | 4 | 19.6 | 18.0 | 7.15 | 51.08 | 0.0 | 58.0 |
| | 5 | 17.9 | 17.0 | 7.48 | 55.93 | 0.0 | 58.0 |

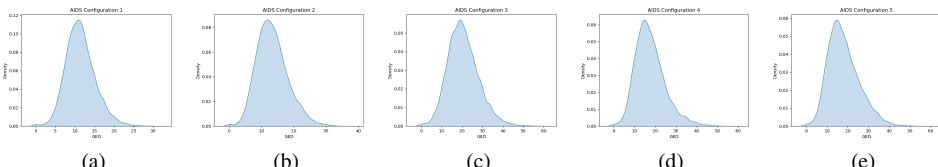

(a)    (b)    (c)    (d)    (e)

Figure 10: Statistics of the AIDS dataset.

Table 8: Statistics of the MUTAG dataset used in the GED approximation.

| DATASET | CONFIGURATIONS | MEAN | MEDIAN | STD | VARIANCE | MIN | MAX |
|---|---|---|---|---|---|---|---|
| MUTAG (MORRIS ET AL., 2020) | 1 | 30.9 | 31.0 | 8.90 | 79.24 | 0.0 | 51.0 |
| | 2 | 15.9 | 16.0 | 4.47 | 20.01 | 0.0 | 26.0 |
| | 3 | 19.3 | 20.0 | 5.33 | 28.39 | 0.0 | 32.0 |
| | 4 | 22.3 | 22.0 | 6.65 | 44.22 | 0.0 | 40.0 |
| | 5 | 20.0 | 20.0 | 5.74 | 32.97 | 0.0 | 37.0 |

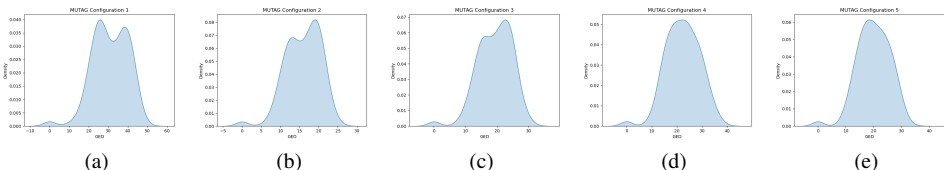

(a)    (b)    (c)    (d)    (e)

Figure 11: Statistics of the MUTAG dataset.

Table 9: Statistics of the PTC MR dataset used in the GED approximation.

| DATASET | CONFIGURATIONS | MEAN | MEDIAN | STD | VARIANCE | MIN | MAX |
|---|---|---|---|---|---|---|---|
| PTC MR (MORRIS ET AL., 2020) | 1 | 13.0 | 13.0 | 5.29 | 28.00 | 0.0 | 30.0 |
| | 2 | 15.3 | 15.0 | 6.61 | 43.80 | 0.0 | 38.0 |
| | 3 | 21.3 | 21.0 | 9.26 | 85.69 | 0.0 | 53.0 |
| | 4 | 19.6 | 18.0 | 9.92 | 98.55 | 0.0 | 60.0 |
| | 5 | 19.5 | 18.0 | 10.0 | 100.9 | 0.0 | 60.0 |

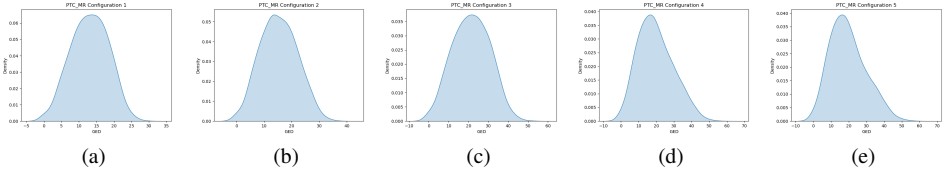

(a)    (b)    (c)    (d)    (e)

Figure 12: Statistics of the PTC MR dataset.

## D.2 Results For Each Configuration

In Tab. 10, 11, 12, 13, 14, we present the results obtained from each model on the three datasets. All results are obtained by cross-validation with 5-fold K-fold scheme, using the 10% of training set as validation set for model selection. For each fold, we select the model with the best performance on the validation set and then use it for inference on the test set.

We optimize each model by searching for configuration settings tailored to each dataset, varying both depth and number of parameters to balance performance quality and execution time. Although this optimization is not the primary focus of our work, it provides a useful estimate of how our approach diverges from prior literature on this specific task.

The only model not subject to parameter-level optimization is EUGENE (Bommakanti et al., 2024). This is because it does not offer control over parameters beyond cost settings, which we follow exactly as specified by the original authors. Another limitation of EUGENE (Bommakanti et al., 2024) is the inability to assign non-uniform costs to edges: the code defines a single edge cost, without differentiating between insertion and deletion, unlike nodes, for which separate costs can be specified for insertion, substitution, and deletion. To address this, we use an average cost, which prevents a fully comparable evaluation with the last two configurations.

Despite this, we observe that EUGENE (Bommakanti et al., 2024) remains, on average, less accurate than GEDAN even in configurations where cost settings are consistent. Furthermore, as shown later, EUGENE (Bommakanti et al., 2024) requires substantially more time for prediction. For these three reasons, summarized in Table 2, we prefer the GEDAN configuration with $\lambda = 1$.

A comparative analysis across different configurations confirms the absence of a universally superior model, in agreement with the no-free-lunch theorem (Wolpert & Macready, 1997). The best performance is highlighted in boldface, while the second-best is underlined. For EUGENE (Bommakanti et al., 2024), we report the standard derivation, as we applied folding to accelerate the approximation using fewer graphs. In contrast, for the approximation proposed by Fischer et al. (2017), we approximated the entire dataset directly, since this method is efficient and, like EUGENE, does not require any training.

Table 10: GED Approximation with Configuration 1.

| | MODEL | AIDS | | | MUTAG | | | PTC MR | | |
| --- | --- | --- | --- | --- | --- | --- | --- | --- | --- | --- |
| | | RMSE ($\downarrow$) | $\tau$ ($\uparrow$) | $\rho$ ($\uparrow$) | RMSE ($\downarrow$) | $\tau$ ($\uparrow$) | $\rho$ ($\uparrow$) | RMSE ($\downarrow$) | $\tau$ ($\uparrow$) | $\rho$ ($\uparrow$) |
| S | SIMGNN [1] | $3.47 \pm 0.33$ | $\mathbf{0.571 \pm 0.05}$ | $\mathbf{0.764 \pm 0.04}$ | $7.96 \pm 1.04$ | $0.562 \pm 0.06$ | $0.802 \pm 0.03$ | $4.08 \pm 0.37$ | $\underline{0.708 \pm 0.04}$ | $\underline{0.884 \pm 0.03}$ |
| | EGSC [47] | $\underline{2.94 \pm 0.12}$ | $0.463 \pm 0.02$ | $0.638 \pm 0.03$ | $5.56 \pm 0.38$ | $0.553 \pm 0.03$ | $0.754 \pm 0.04$ | $\underline{2.37 \pm 0.21}$ | $\mathbf{0.729 \pm 0.02}$ | $\mathbf{0.899 \pm 0.03}$ |
| | GREED [48] | $3.10 \pm 0.20$ | $0.349 \pm 0.06$ | $0.545 \pm 0.06$ | $5.66 \pm 0.23$ | $0.515 \pm 0.07$ | $0.705 \pm 0.09$ | $\mathbf{2.18 \pm 0.08}$ | $0.666 \pm 0.06$ | $0.849 \pm 0.04$ |
| | ERIC [74] | $3.66 \pm 0.34$ | $0.425 \pm 0.05$ | $0.656 \pm 0.03$ | $8.65 \pm 1.28$ | $0.525 \pm 0.06$ | $0.775 \pm 0.06$ | $4.32 \pm 1.10$ | $0.706 \pm 0.04$ | $0.865 \pm 0.04$ |
| | GRAPHEDX$_{\text{XOR}}$ [29] | $3.58 \pm 0.46$ | $\underline{0.557 \pm 0.04}$ | $\underline{0.745 \pm 0.08}$ | $\underline{5.33 \pm 0.14}$ | $0.605 \pm 0.02$ | $0.868 \pm 0.01$ | $3.30 \pm 0.53$ | $0.651 \pm 0.06$ | $0.835 \pm 0.06$ |
| | GEDAN($\lambda = 0$) | $\mathbf{2.82 \pm 0.19}$ | $0.484 \pm 0.05$ | $0.694 \pm 0.05$ | $\mathbf{5.20 \pm 0.15}$ | $\mathbf{0.650 \pm 0.04}$ | $\mathbf{0.887 \pm 0.01}$ | $3.18 \pm 0.65$ | $0.655 \pm 0.08$ | $0.826 \pm 0.07$ |
| U | APPROXIMATED GED [21] | $\underline{3.89}$ | $0.372$ | $0.542$ | $\underline{16.46}$ | $\underline{0.560}$ | $\underline{0.757}$ | $\mathbf{3.49}$ | $\underline{0.630}$ | $\underline{0.797}$ |
| | EUGENE [6] | $4.32 \pm 0.16$ | $\mathbf{0.505 \pm 0.03}$ | $\mathbf{0.658 \pm 0.05}$ | $18.68 \pm 0.80$ | $0.507 \pm 0.09$ | $0.684 \pm 0.07$ | $5.15 \pm 0.76$ | $0.625 \pm 0.06$ | $0.763 \pm 0.05$ |
| | GEDAN($\lambda = 1$) | $\mathbf{3.54 \pm 0.14}$ | $\underline{0.437 \pm 0.06}$ | $\underline{0.652 \pm 0.06}$ | $\mathbf{13.30 \pm 0.85}$ | $\mathbf{0.611 \pm 0.09}$ | $\mathbf{0.852 \pm 0.04}$ | $\underline{3.51 \pm 0.31}$ | $\mathbf{0.649 \pm 0.07}$ | $\mathbf{0.823 \pm 0.05}$ |

Table 11: GED Approximation with Configuration 2.

| | MODEL | AIDS | | | MUTAG | | | PTC MR | | |
| --- | --- | --- | --- | --- | --- | --- | --- | --- | --- | --- |
| | | RMSE ($\downarrow$) | $\tau$ ($\uparrow$) | $\rho$ ($\uparrow$) | RMSE ($\downarrow$) | $\tau$ ($\uparrow$) | $\rho$ ($\uparrow$) | RMSE ($\downarrow$) | $\tau$ ($\uparrow$) | $\rho$ ($\uparrow$) |
| S | SIMGNN [1] | $5.82 \pm 0.36$ | $\mathbf{0.591 \pm 0.03}$ | $\mathbf{0.810 \pm 0.02}$ | $6.02 \pm 0.76$ | $\mathbf{0.631 \pm 0.04}$ | $0.821 \pm 0.04$ | $4.08 \pm 0.51$ | $0.755 \pm 0.04$ | $0.915 \pm 0.03$ |
| | EGSC [47] | $\underline{3.36 \pm 0.11}$ | $0.512 \pm 0.02$ | $0.696 \pm 0.02$ | $\mathbf{2.47 \pm 0.21}$ | $\underline{0.604 \pm 0.03}$ | $0.808 \pm 0.03$ | $\underline{2.52 \pm 0.19}$ | $\mathbf{0.782 \pm 0.02}$ | $\mathbf{0.931 \pm 0.02}$ |
| | GREED [48] | $\mathbf{2.92 \pm 0.09}$ | $0.334 \pm 0.07$ | $0.571 \pm 0.07$ | $4.48 \pm 0.86$ | $0.585 \pm 0.04$ | $0.676 \pm 0.06$ | $\mathbf{2.06 \pm 0.18}$ | $0.711 \pm 0.05$ | $0.881 \pm 0.05$ |
| | ERIC [74] | $5.25 \pm 0.42$ | $0.545 \pm 0.03$ | $\underline{0.763 \pm 0.05}$ | $5.78 \pm 0.35$ | $0.588 \pm 0.07$ | $\mathbf{0.831 \pm 0.03}$ | $4.94 \pm 0.89$ | $0.763 \pm 0.03$ | $0.911 \pm 0.02$ |
| | GRAPHEDX$_{\text{XOR}}$ [29] | $4.87 \pm 0.10$ | $\underline{0.557 \pm 0.06}$ | $0.745 \pm 0.05$ | $3.85 \pm 0.51$ | $0.574 \pm 0.08$ | $0.759 \pm 0.07$ | $3.12 \pm 0.22$ | $0.694 \pm 0.07$ | $0.833 \pm 0.06$ |
| | GEDAN($\lambda = 0$) | $3.52 \pm 0.14$ | $0.511 \pm 0.04$ | $0.721 \pm 0.04$ | $\underline{3.36 \pm 0.49}$ | $0.587 \pm 0.09$ | $\underline{0.829 \pm 0.04}$ | $2.85 \pm 0.47$ | $\underline{0.770 \pm 0.02}$ | $\underline{0.917 \pm 0.02}$ |
| U | APPROXIMATED GED [21] | $6.29$ | $0.366$ | $0.532$ | $\underline{8.01}$ | $\underline{0.442}$ | $\underline{0.589}$ | $\underline{4.42}$ | $0.661$ | $0.829$ |
| | EUGENE [6] | $\underline{5.39 \pm 0.19}$ | $\mathbf{0.514 \pm 0.02}$ | $\underline{0.691 \pm 0.04}$ | $7.02 \pm 1.37$ | $0.402 \pm 0.18$ | $0.588 \pm 0.13$ | $5.14 \pm 0.54$ | $\underline{0.709 \pm 0.03}$ | $\underline{0.848 \pm 0.02}$ |
| | GEDAN($\lambda = 1$) | $\mathbf{3.91 \pm 0.09}$ | $\underline{0.482 \pm 0.06}$ | $\mathbf{0.698 \pm 0.05}$ | $\mathbf{6.34 \pm 0.80}$ | $\mathbf{0.595 \pm 0.11}$ | $\mathbf{0.813 \pm 0.05}$ | $\mathbf{3.44 \pm 0.24}$ | $\mathbf{0.742 \pm 0.05}$ | $\mathbf{0.897 \pm 0.03}$ |

Table 12: GED Approximation with Configuration 3.

| | MODEL | AIDS | | | MUTAG | | | PTC MR | | |
|---|---|---|---|---|---|---|---|---|---|---|
| | | RMSE (↓) | τ (↑) | ρ (↑) | RMSE (↓) | τ (↑) | ρ (↑) | RMSE (↓) | τ (↑) | ρ (↑) |
| S | SimGNN [1] | 7.32 ± 0.91 | 0.523 ± 0.05 | 0.706 ± 0.04 | 6.55 ± 0.23 | 0.513 ± 0.05 | 0.722 ± 0.03 | 6.84 ± 1.05 | 0.605 ± 0.06 | 0.767 ± 0.04 |
| | EGSC [47] | 6.15 ± 0.16 | 0.413 ± 0.01 | 0.579 ± 0.01 | **2.96 ± 0.13** | 0.572 ± 0.04 | 0.784 ± 0.03 | 5.03 ± 0.21 | 0.681 ± 0.02 | **0.860 ± 0.02** |
| | GREED [48] | **3.05 ± 0.16** | 0.467 ± 0.05 | 0.663 ± 0.06 | 4.96 ± 0.78 | **0.592 ± 0.05** | 0.659 ± 0.06 | **2.85 ± 0.34** | 0.676 ± 0.07 | 0.847 ± 0.06 |
| | ERIC [74] | 6.74 ± 0.53 | 0.438 ± 0.04 | 0.631 ± 0.05 | 5.70 ± 0.31 | 0.560 ± 0.05 | 0.793 ± 0.04 | 6.11 ± 1.12 | 0.630 ± 0.05 | 0.795 ± 0.04 |
| | GraphEdX$_{XOR}$ [29] | 5.45 ± 0.43 | **0.591 ± 0.02** | **0.811 ± 0.02** | 3.98 ± 0.65 | 0.571 ± 0.08 | 0.753 ± 0.09 | 5.10 ± 0.21 | **0.695 ± 0.02** | 0.848 ± 0.03 |
| | GEDAN(λ = 0) | 5.88 ± 0.28 | 0.426 ± 0.05 | 0.641 ± 0.04 | 4.08 ± 0.56 | 0.582 ± 0.08 | **0.825 ± 0.04** | 5.42 ± 1.03 | 0.664 ± 0.07 | 0.835 ± 0.06 |
| U | Approximated GED [21] | 8.15 | 0.328 | 0.495 | 8.01 | 0.444 | 0.613 | 6.30 | **0.643** | 0.804 |
| | EUGENE [6] | 12.50 ± 0.94 | **0.437 ± 0.04** | 0.564 ± 0.05 | 9.18 ± 1.56 | 0.372 ± 0.20 | 0.553 ± 0.15 | 10.85 ± 0.99 | 0.628 ± 0.07 | 0.764 ± 0.05 |
| | GEDAN(λ = 1) | **6.72 ± 0.18** | 0.398 ± 0.05 | **0.608 ± 0.04** | 15.12 ± 1.18 | **0.569 ± 0.08** | **0.830 ± 0.03** | **6.29 ± 0.40** | 0.639 ± 0.06 | **0.815 ± 0.05** |

Table 13: GED Approximation with Configuration 4.

| | MODEL | AIDS | | | MUTAG | | | PTC MR | | |
|---|---|---|---|---|---|---|---|---|---|---|
| | | RMSE (↓) | τ (↑) | ρ (↑) | RMSE (↓) | τ (↑) | ρ (↑) | RMSE (↓) | τ (↑) | ρ (↑) |
| S | SimGNN [1] | 7.05 ± 0.72 | 0.525 ± 0.04 | 0.755 ± 0.04 | 6.31 ± 0.66 | 0.641 ± 0.02 | 0.833 ± 0.03 | 7.11 ± 0.58 | 0.658 ± 0.04 | 0.825 ± 0.06 |
| | EGSC [47] | 5.21 ± 0.17 | 0.474 ± 0.03 | 0.651 ± 0.03 | **3.27 ± 0.16** | **0.672 ± 0.02** | **0.863 ± 0.02** | 4.89 ± 0.62 | **0.709 ± 0.02** | **0.880 ± 0.02** |
| | GREED [48] | **3.06 ± 0.22** | 0.484 ± 0.07 | 0.698 ± 0.06 | 4.77 ± 0.67 | 0.596 ± 0.04 | 0.648 ± 0.04 | **3.25 ± 0.19** | 0.696 ± 0.05 | 0.858 ± 0.03 |
| | ERIC [74] | 6.17 ± 0.55 | 0.507 ± 0.04 | 0.728 ± 0.05 | 5.54 ± 0.65 | 0.581 ± 0.06 | 0.652 ± 0.06 | 5.81 ± 1.10 | 0.652 ± 0.08 | 0.815 ± 0.04 |
| | GraphEdX$_{XOR}$ [29] | 4.65 ± 0.32 | **0.605 ± 0.03** | **0.835 ± 0.02** | 4.44 ± 0.65 | 0.624 ± 0.04 | 0.838 ± 0.05 | 5.25 ± 0.68 | 0.666 ± 0.05 | 0.861 ± 0.05 |
| | GEDAN(λ = 0) | 5.17 ± 0.26 | 0.479 ± 0.04 | 0.718 ± 0.04 | 4.82 ± 0.58 | 0.599 ± 0.10 | 0.825 ± 0.06 | 5.27 ± 1.04 | 0.681 ± 0.08 | 0.856 ± 0.06 |
| U | Approximated GED [21] | 8.22 | 0.181 | 0.269 | 9.24 | 0.278 | 0.416 | 10.96 | 0.290 | 0.350 |
| | EUGENE [6] | 9.50 ± 0.56 | **0.449 ± 0.03** | 0.659 ± 0.04 | 10.96 ± 1.43 | 0.467 ± 0.13 | 0.671 ± 0.11 | 9.54 ± 1.26 | 0.622 ± 0.09 | 0.796 ± 0.07 |
| | GEDAN(λ = 1) | **5.77 ± 0.12** | 0.432 ± 0.04 | **0.675 ± 0.04** | **7.50 ± 0.72** | **0.604 ± 0.10** | **0.827 ± 0.06** | **5.75 ± 0.52** | **0.638 ± 0.06** | **0.831 ± 0.04** |

Table 14: GED Approximation with Configuration 5.

| | MODEL | AIDS | | | MUTAG | | | PTC MR | | |
|---|---|---|---|---|---|---|---|---|---|---|
| | | RMSE (↓) | τ (↑) | ρ (↑) | RMSE (↓) | τ (↑) | ρ (↑) | RMSE (↓) | τ (↑) | ρ (↑) |
| S | SimGNN [1] | 5.98 ± 0.63 | 0.571 ± 0.06 | 0.788 ± 0.05 | 4.77 ± 0.85 | 0.656 ± 0.04 | 0.837 ± 0.03 | 4.64 ± 0.55 | 0.718 ± 0.03 | 0.885 ± 0.03 |
| | EGSC [47] | 5.47 ± 0.49 | 0.479 ± 0.02 | 0.653 ± 0.03 | **2.76 ± 0.16** | **0.667 ± 0.02** | **0.864 ± 0.02** | 4.50 ± 0.35 | **0.731 ± 0.04** | **0.899 ± 0.03** |
| | GREED [48] | **3.14 ± 0.22** | 0.509 ± 0.06 | 0.690 ± 0.07 | 4.51 ± 0.40 | 0.630 ± 0.03 | 0.697 ± 0.04 | **2.99 ± 0.46** | 0.719 ± 0.07 | 0.872 ± 0.05 |
| | ERIC [74] | 5.36 ± 0.43 | 0.461 ± 0.04 | 0.666 ± 0.03 | 4.56 ± 0.77 | 0.582 ± 0.03 | 0.805 ± 0.03 | 5.05 ± 0.83 | 0.687 ± 0.04 | 0.865 ± 0.05 |
| | GraphEdX$_{XOR}$ [29] | 4.65 ± 0.48 | **0.605 ± 0.02** | **0.835 ± 0.02** | 4.44 ± 0.76 | 0.624 ± 0.07 | 0.838 ± 0.03 | 5.25 ± 0.32 | 0.666 ± 0.03 | 0.861 ± 0.04 |
| | GEDAN(λ = 0) | 5.50 ± 0.25 | 0.486 ± 0.04 | 0.705 ± 0.04 | 4.91 ± 0.47 | 0.521 ± 0.06 | 0.755 ± 0.03 | 5.51 ± 1.10 | 0.693 ± 0.06 | 0.848 ± 0.07 |
| U | Approximated GED [21] | 9.42 | 0.116 | 0.176 | 10.44 | 0.285 | 0.407 | 11.96 | 0.250 | 0.277 |
| | EUGENE [6] | 11.27 ± 0.31 | 0.260 ± 0.03 | 0.333 ± 0.05 | 9.91 ± 1.55 | 0.354 ± 0.14 | 0.523 ± 0.12 | 12.27 ± 1.09 | 0.379 ± 0.09 | 0.423 ± 0.08 |
| | GEDAN(λ = 1) | **6.00 ± 0.10** | **0.433 ± 0.04** | **0.658 ± 0.04** | **8.65 ± 0.94** | **0.513 ± 0.05** | **0.741 ± 0.02** | **5.56 ± 0.55** | **0.650 ± 0.06** | **0.839 ± 0.05** |

## D.3 ADDITIONAL RESULTS ON LARGE GRAPHS

One of the key points raised concerns the possibility of adopting an unsupervised approach to approximate the *Graph Edit Distance* (GED). As highlighted by Neuhaus et al. (2006a), increasing the graph size inevitably reduces accuracy, since the problem is NP-hard and approximations are required, which in turn lead to suboptimal solutions.

Similarly, in our setting, larger graphs make it increasingly difficult to identify the optimal matching, as the number of candidate pairs grows with the number of nodes. To assess scalability, we consider graphs of up to 32 nodes from the BZR dataset (Morris et al., 2020) using a $64 \times 64$ model, graphs of up to 64 nodes from the Mutagenicity and NCI1 datasets with a $128 \times 128$ model, and finally the Synthie dataset (average 95 nodes) with a $256 \times 256$ model. Despite relying on relatively few graphs due to the training cost, our approach remains consistently faster than EUGENE (Bommakanti et al., 2024).

To estimate GED, we adopt the algorithm proposed by Fischer et al. (2017), implemented in the GMATCH4PY library, and also report the *GED baseline*, a strict lower bound defined as the absolute difference between the number of nodes and edges in the graphs.

Since our model trains more efficiently, we allow it to run for a sufficient number of epochs so that its overall training cost matches the prediction cost of EUGENE (Bommakanti et al., 2024), thereby ensuring a fair comparison. In cases where the number of nodes is particularly large, EUGENE (Bommakanti et al., 2024) failed to provide results within a reasonable time; in such scenarios, we retain the comparison with the method of Fischer et al. (2017).

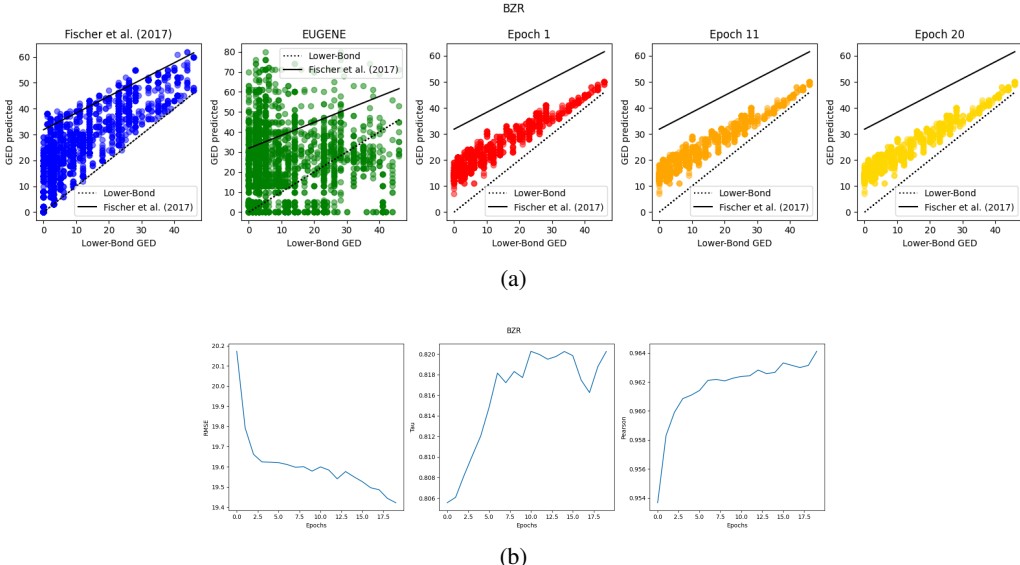

Figure 13: (a) Comparison with EUGENE and Fischer et al. (2017). The black line indicates the interpolation of Fischer et al. (2017) maxima. (b) Training dynamics of GEDAN, showing the loss, $\tau$, and $\rho$. Training for 20 epochs requires 27 minutes 5 seconds, compared to 26 minutes 21 seconds for EUGENE (Bommakanti et al., 2024).

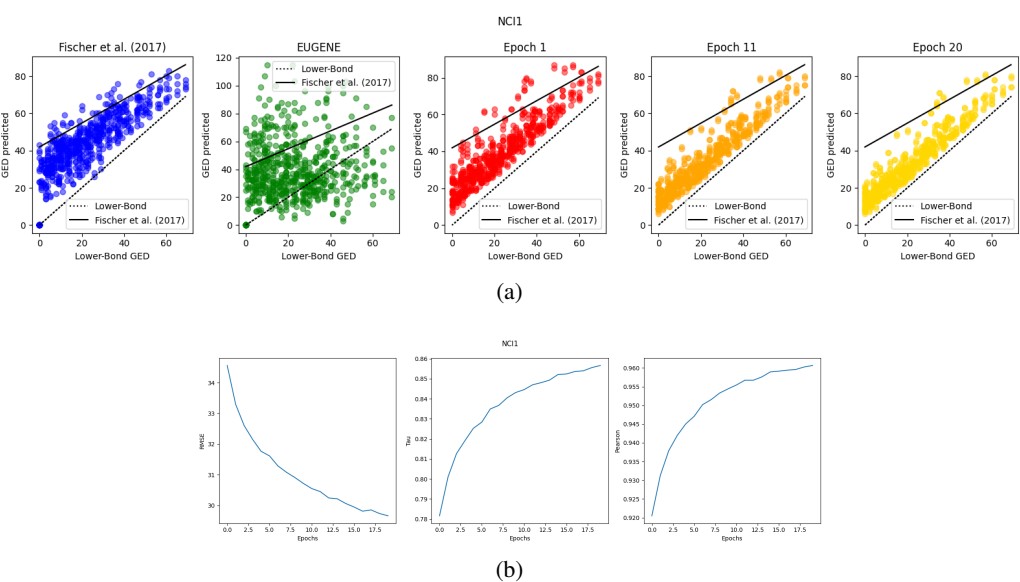

Figure 14: (a) Comparison with EUGENE and Fischer et al. (2017). The black line indicates the interpolation of Fischer et al. (2017) maxima. (b) Training dynamics of GEDAN, showing the loss, $\tau$, and $\rho$. Training for 20 epochs requires 6 minutes 21 seconds, compared to 6 minutes 44 seconds for EUGENE.

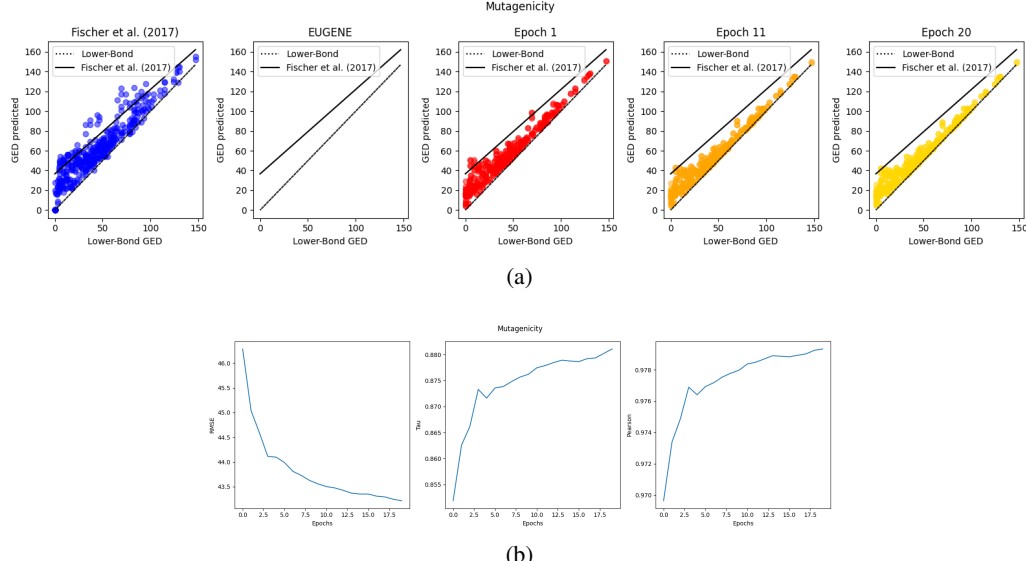

Figure 15: (a) Comparison with Fischer et al. (2017). The black line indicates the interpolation of Fischer et al. (2017) maxima. (b) Training dynamics of GEDAN, showing the loss, $\tau$, and $\rho$. Training for 20 epochs requires 2 minutes 56 seconds.

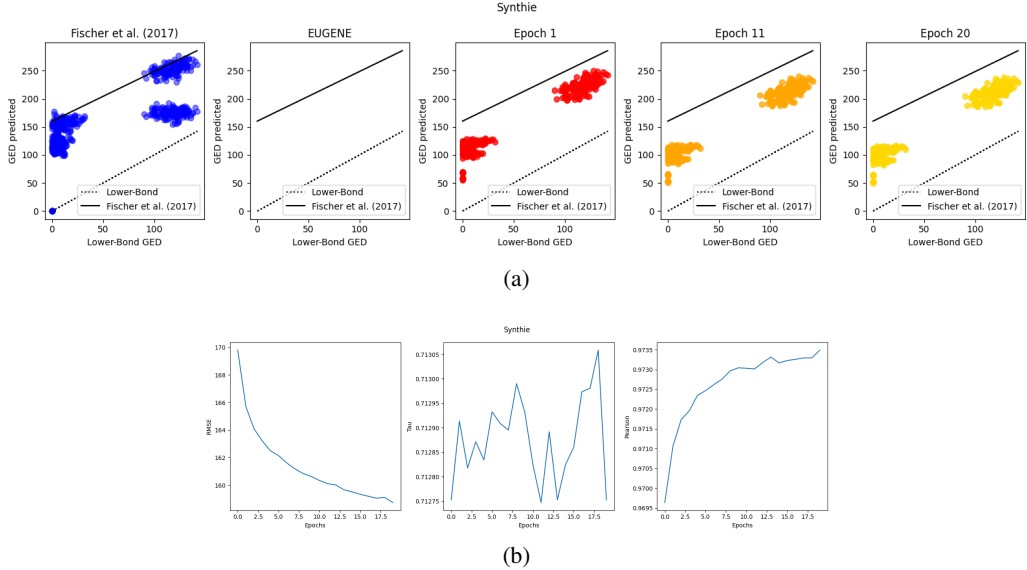

Figure 16: (a) Comparison with Fischer et al. (2017). The black line indicates the interpolation of Fischer et al. (2017) maxima. (b) Training dynamics of GEDAN, showing the loss, $\tau$, and $\rho$. Training for 20 epochs requires 5 minutes 4 seconds.

# E DETAILS OF EDIT COST OPTIMIZATION

In this section, we provide additional details on the learning of edit costs. We illustrate the flexibility of the learnable cost functions employed by GEDAN, describe the datasets and loss functions used, and report further details on the results that are not covered in the main paper.

### E.1 Details on Edit Cost Datasets

For edit cost optimization we use two molecular datasets, varying the size and type of task required. The graphs are limited to 32 nodes, thus using the Gumbel-Sinkhorn $64 \times 64$ module, with the performance reported in Appendix C. We show the characteristics of the datasets used in Table 15.

Table 15: Characteristics of the dataset used in edit cost optimization.

| FreeSolv (Mobley & Guthrie, 2014) | | | BBBP (Martins et al., 2012) | | |
|---|---|---|---|---|---|
| N. graphs | Avg. nodes | Avg. edges | N. graphs | Avg. nodes | Avg. edges |
| 642 | 8.72 | 16.78 | 1734 | 20.84 | 44.88 |

FreeSolv (Mobley & Guthrie, 2014) is a small dataset that collects hydration free energies (HFE), both experimental and calculated, for small molecules in water. It is commonly used for the prediction of molecular properties in the field of computational chemistry. The choice of this dataset is due to its small size, especially in application contexts one does not have a lot of data, so we verified that the model would work in this context.

Blood-Brain Barrier Penetration (BBBP) (Martins et al., 2012) is dataset that models the permeability of the blood-brain barrier, which can prevent the passage of the of drugs, hormones and neurotransmitters. The dataset tests the ability of a molecule to cross this barrier with binary labels. The dataset is not large to test the model in a real scenario of experimental data, where there are often not many.

We introduce a new dataset, the GPU Network Cost Dataset (GNCD), specifically designed for our experiments. The dataset consists of synthetic graphs in which each node represents a GPU compute node, associated with its purchase cost, and each edge represents a cable required to connect two GPUs. The total value assigned to a graph corresponds to the overall cost of building the network, capturing the intrinsic cost asymmetry of the domain: GPU nodes are significantly more expensive than connection cables.

In our experimental setup, GNCD serves as a strongly asymmetric dataset for studying graph transformation costs. We define the transformation cost between two graphs as the cost of the components that appear in one graph but not in the other, with one important convention: if $G_1 \subseteq G_2$, then the cost of transforming $G_2$ into $G_1$ is zero, since all components needed to obtain $G_1$ are already contained in $G_2$. This convention introduces a marked asymmetry not only between the costs associated with nodes and edges, but also between addition and deletion operations: deletion is considered free, whereas addition incurs a significant cost. We employ GNCD in the experiments reported in Section E.3.

The dataset contains 200 unique graphs, with an average of 13.2 nodes and 55.8 edges per graph. We include five different types of GPU computation nodes, priced at 2,400, 5,000, 6,000, 15,000, and 25,000 cost units, while edges (cables) cost only 25 units.

### E.2 Loss and Training Strategy

The training of GEDAN, aimed at aligning the GED with the functional distance and thus learning the optimal set of edit costs, relies on the joint optimization of two loss functions:

- a contrastive loss based on the softmax applied to GED ,
- and a loss for the downstream task (e.g., regression or classification).

During training, for each query we select $H$ keys from the training set by causal sampling, ensuring semantic consistency with the target to be predicted. Of these $H$ keys, the first and last are fixed while the remaining $H - 2$ are selected according to the nature of the task.

In detail:

- For regression, the fixed keys correspond to the instances with minimum and maximum target value; the middle keys are selected so that their targets are ordered with respect to that of the query.

- For binary classification, $(H - 2)/2$ instances are selected for each class. The two fixed keys represent one instance of the negative class and one instance of the positive class.

This strategy ensures a balanced and semantically consistent distribution, although it does not include hard negative sampling, which may make optimization less effective in ambiguous cases. The introduction of more sophisticated sampling techniques is left for future work.

Contrastive loss, applied to GED distances, is defined as:

$$\mathcal{L}_i^{\text{contr}} = -\log\left(\frac{\exp\left(-\text{GED}_{i,y_i}/T\right)}{\sum_{j=1}^{H} \exp\left(-\text{GED}_{i,j}/T\right)}\right) \tag{21}$$

where:

- $\text{GED}_{i,j}$ is the distance between the query $i$ and the key $j$,
- $y_i \in \{1, \ldots, H\}$ is the index of the positive key,
- $T \in \mathbb{R}^+$ is the temperature hyperparameter of the softmax distribution.

This formulation can be interpreted as a differentiable approximation of minimizing the distance between the query and its assigned positive key, while simultaneously penalizing the negative keys proportionally to their similarity (i.e., inverse of GED). As a result, the learned distribution encourages alignment with semantically coherent keys and drives the model to build discriminative representations.

The assignment of the positive key $y_i$ depends on the nature of the task:

- For regression, the positive key is the one with the closest label value to the query value, i.e.:

$$y_i = \arg\min_j \left| y_i^{\text{true}} - y_j^{\text{key}} \right| \tag{22}$$

  ensuring that the model focuses on neighboring keys in the continuous target domain.

- For classification, a key belonging to the same class as the query is selected. In the presence of multiple candidate keys with the same class, the average GED distance of multiple candidate keys to the query is selected. This approach promotes a more robust semantic match, avoiding outliers.

The loss associated with the downstream task is computed using a separate MLP that takes as input the set of $\text{GED}_{i,j}$ values. Since semantic coherence between the query and keys has been preserved through the sampling strategy, we can directly optimize with respect to the query's label as follows:

- For the regression:

$$\mathcal{L}_i^{\text{reg}} = \left\| \hat{y}_i - y_i^{\text{true}} \right\|^2 \tag{23}$$

- For binary classification:

$$\mathcal{L}_i^{\text{cls}} = -y_i^{\text{true}} \log \hat{y}_i - (1 - y_i^{\text{true}}) \log(1 - \hat{y}_i) \tag{24}$$

Finally, the total loss is obtained as a weighted combination of the two components:

$$\mathcal{L} = \frac{1}{B} \sum_{i=1}^{B} \left( \mathcal{L}_i^{\text{contr}} + \lambda \cdot \mathcal{L}_i^{\text{task}} \right) \tag{25}$$

where:

- $B$ is the batch size,
- $\mathcal{L}_i^{\text{task}} \in \{\mathcal{L}_i^{\text{reg}}, \mathcal{L}_i^{\text{cls}}\}$ depending on the task,
- $\lambda \in \mathbb{R}^+$ It is a hyperparameter that balances the two components.

E.3    LEARNING EDIT COSTS ACROSS LEVELS OF EXPRESSIVENESS

As defined by Formula 8 in GEDAN, cost learning is highly flexible, as it depends on the choice and implementation of the functions $f_\theta^{(k)}$. In this section, we analyze how GEDAN learns edit costs using datasets for which ground-truth edit costs are known. This setting simplifies the analysis compared to molecular datasets – where no such ground truth exists – and allows us to clearly illustrate how a framework like GEDAN can adapt to different domains. We examine three cost-function configurations:

1. Generic costs, independent of node type.

2. Node-specific costs combined with a linear function $f_\theta^{(k)}$, yielding a scalar-weighted sum that trades off expressiveness for linearity.

3. The original non-linear formulation combined with the GAM.

We conduct our analysis on three datasets. For MUTAG, we consider configurations 4 and 5 both asymmetric (see Table 6). In configuration 4, node and edge insertions cost twice as much as deletions, whereas in configuration 5 the opposite holds. The GNCD dataset, instead, is a strongly asymmetric collection of GPU-computation graphs, where deletions are free and insertions incur a cost. Node-specific GPU prices range from 2.4k to 25k units, and edges cost 25 units, providing a setting with heterogeneous node-dependent edit costs.

We begin with the generic-cost setting, where the functions $f_\theta^{(k)}$ are disabled. In this case, GEDAN can only learn the global edit costs $a^\oplus, a^\ominus, b^\oplus, b^\ominus$, without distinguishing node types. In our experiment on MUTAG configurations 4 and 5, we prevent GEDAN from using the matrices $\mathbf{C}_k$, constraining it to learn only the generic edit parameters, which are randomly initialized to identical values across both configurations. Minimizing the training loss therefore forces GEDAN to adjust these generic costs toward their respective theoretical ground-truth values.

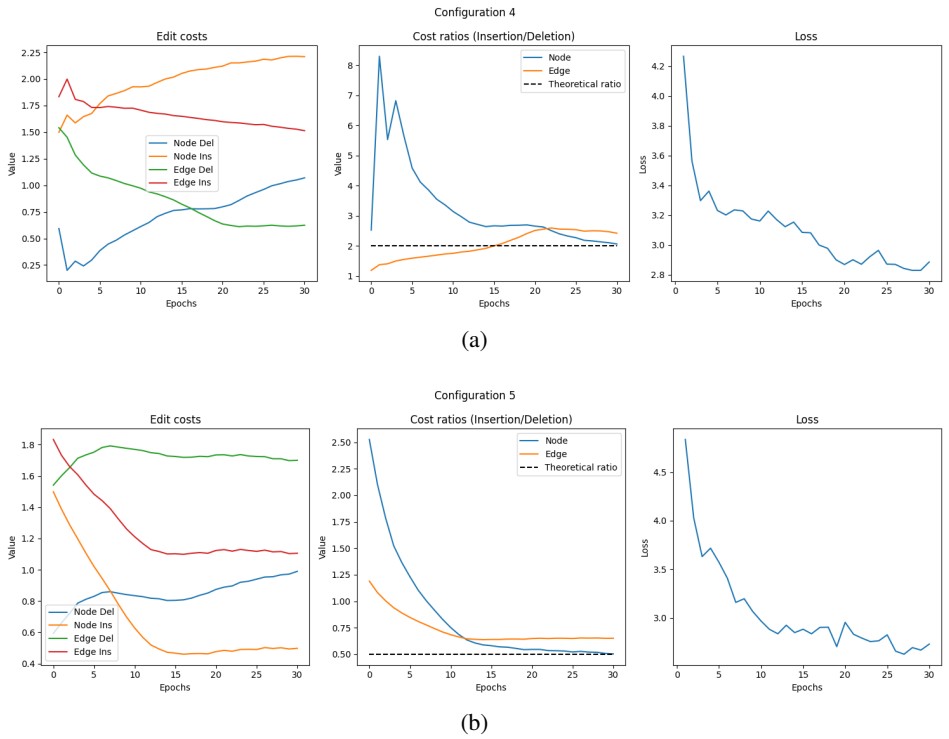

(a)

(b)

Figure 17: Trajectories of the edit costs $a^\oplus, a^\ominus, b^\oplus, b^\ominus$ during optimization on MUTAG.

As shown in Figure 17, tracking the evolution of the costs during training reveals that GEDAN converges to an insertion-deletion ratio of 2:1 for configuration 4 and 1:2 for configuration 5. This

happens despite the identical random initialization of $a^{\oplus}, a^{\ominus}, b^{\oplus}, b^{\ominus}$, and matches the expected theoretical ratios: in configuration 4, node and edge insertions cost twice as much as deletions, whereas in configuration 5 deletions cost twice as much as insertions.

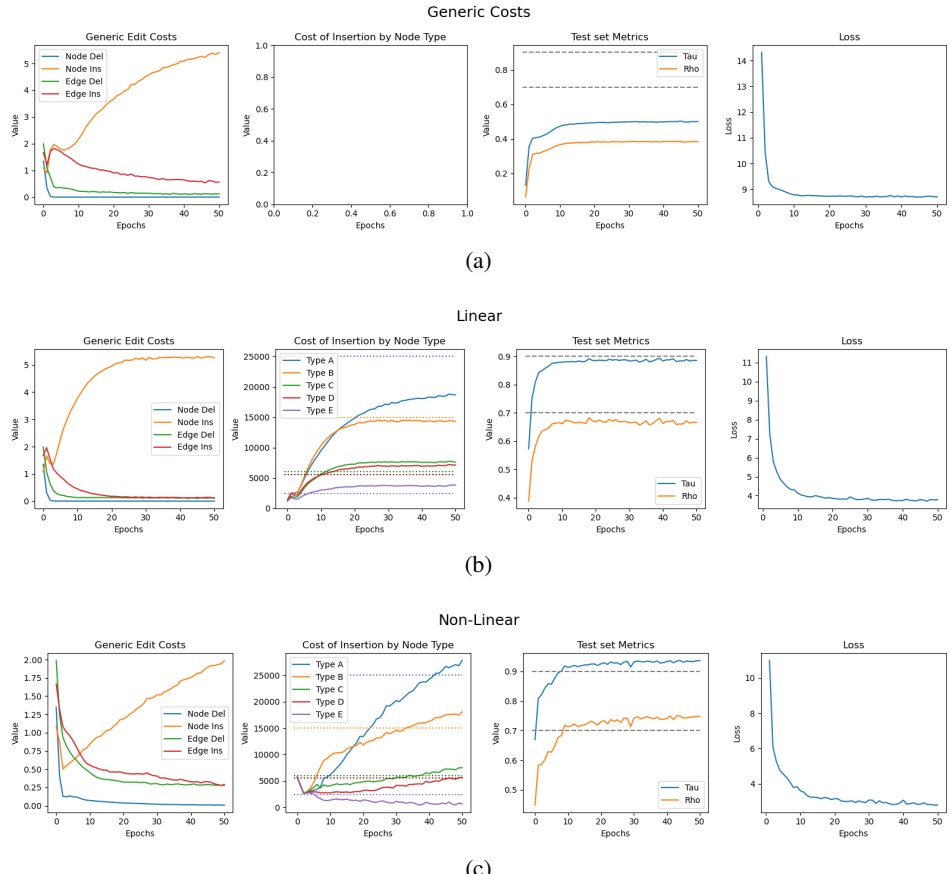

Figure 18: Cost trajectories during optimization on GNCD. (a) generic costs only; (b) linear fine-grained costs; (c) non-linear fine-grained costs.

In the second experiment, we compare the three cost-function configurations on the GNCD dataset. In Figure 18, we show the trajectories of both the generic and the fine-grained substitution costs, focusing specifically on node-insertion costs at the first level, i.e., $f_\theta^0(\epsilon, n_i)$ where $n_i$ is one of the five node types in GNCD. Since ground-truth values are available, we can directly inspect whether the learned functions maintain the correct cost proportions at a finer granularity.

Table 16: Performance of the three GEDAN cost-function variants on MUTAG (conf. 4) and GNCD, showing how expressiveness affects accuracy.

| | | GEDAN EDIT COST | | |
| --- | --- | --- | --- | --- |
| DATASETS | | GENERIC COSTS | SCALAR-WEIGHTED SUM | NON-LINEAR (GAM) |
| MUTAG (CONF. 4) | RMSE ($\downarrow$) | $5.23 \pm 0.99$ | $4.93 \pm 0.69$ | $\mathbf{4.82 \pm 0.58}$ |
| | $\tau$ ($\uparrow$) | $0.484 \pm 0.15$ | $0.514 \pm 0.11$ | $\mathbf{0.599 \pm 0.10}$ |
| | $\rho$ ($\uparrow$) | $0.727 \pm 0.09$ | $0.742 \pm 0.08$ | $\mathbf{0.825 \pm 0.06}$ |
| GNCD (APPENDIX E.1) | NRMSE % ($\downarrow$) | $11.52 \pm 1.25$ | $6.32 \pm 0.67$ | $\mathbf{4.96 \pm 0.41}$ |
| | $\tau$ ($\uparrow$) | $0.379 \pm 0.08$ | $0.683 \pm 0.04$ | $\mathbf{0.762 \pm 0.02}$ |
| | $\rho$ ($\uparrow$) | $0.521 \pm 0.07$ | $0.868 \pm 0.03$ | $\mathbf{0.944 \pm 0.02}$ |

As shown in Table 16, the choice of cost function directly affects model performance. On MUTAG – where the task-specific functional distance $d_f$ is asymmetric but structurally simple – the three variants perform comparably, although the non-linear model consistently yields the best results.

In contrast, GNCD exhibits strong node-type–specific asymmetries, making generic-cost models underfit due to insufficient expressive capacity. Interestingly, all three models converge the generic node deletion cost toward zero during training, which is desirable since deletions are defined as free in GNCD. The main differences arise in the modeling of substitution costs: despite the well-defined cost structure, the linear model fails to represent these values accurately, whereas the non-linear GAM-based variant provides substantially more precise estimates, which is reflected in the better final metrics.

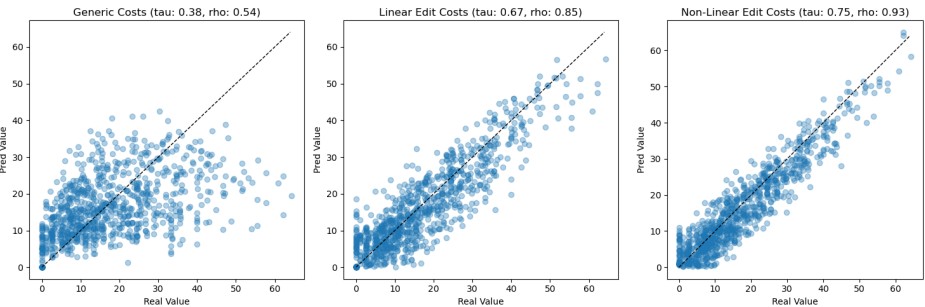

Figure 19: Predicted vs. ground-truth edit costs on GNCD for the three $f_\theta^{(k)}$ (generic, linear scalar-weighted, and non-linear GAM), illustrating how well the learned costs preserve the ordering of the true functional distances $d_f$.

### E.4 EDIT COST OPTIMIZATION EFFECT

In Figs.20 and21, we present the *Principal Component Analysis* (PCA)(Dunteman, 1989) results for the FreeSolv and BBBP datasets, respectively. PCA is applied to the 16 GED embedding values predicted by the networks GEDAN($\lambda = 1$), GMSM ($\epsilon \geq 0$) (Pellizzoni et al., 2024), GMSM ($\epsilon = 0$) (Pellizzoni et al., 2024), and GEDAN ($\lambda = 0$). All models employ 5 layers to ensure that both architectures perceive the same graph depth. For GMSM (Pellizzoni et al., 2024), we use the `retrieve` method, which provides a similarity value used to construct the embeddings, whereas GEDAN directly outputs the GED, which we employ for PCA. For GEDAN ($\lambda = 1$), we adopt the uniform fixed cost variant with costs equal to 1.

In Figs.20 and21, each point corresponds to a graph, and its color indicates the associated label value. PCA clearly illustrates the effect of optimization on the predicted GED: optimized methods generally exhibit smoother gradient transitions, leading to a more accurate alignment with the target values. This additional experiment further demonstrates that the use of uniform costs is often poorly representative for many real-world scenarios.

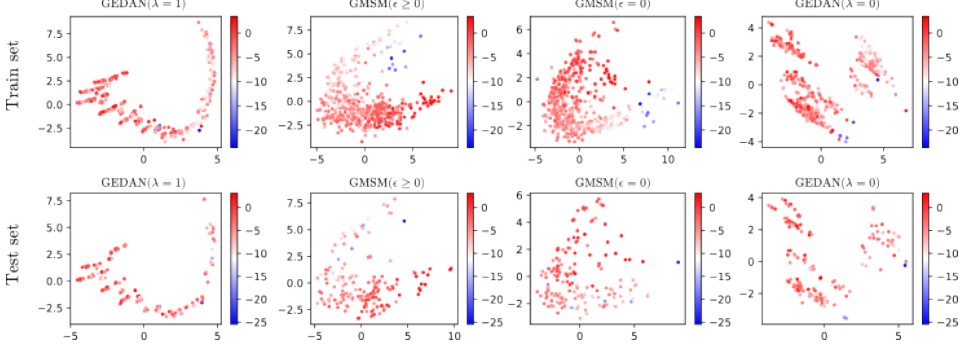

Figure 20: PCA of the FreeSolv dataset (Mobley & Guthrie, 2014).

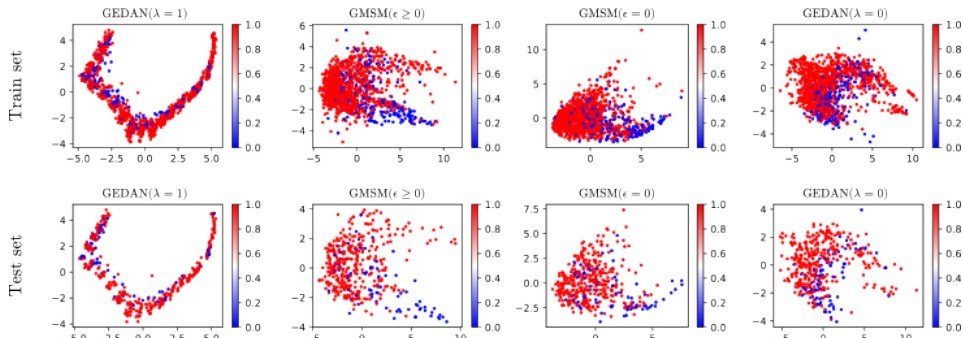

Figure 21: PCA of the BBBP dataset (Martins et al., 2012).

# F    COMPUTATIONAL EFFICIENCY ANALYSIS

As discussed in the main paper, one limitation of our model is its higher computational cost, which results in slower performance compared to state-of-the-art methods. In this section, we analyze the computational efficiency of our approach relative to existing methods, reporting both training and inference times as well as peak memory usage for each model, as summarized in Table 17.

Table 17: Computational efficiency analysis performed on a GeForce RTX 4070 Ti GPU with a batch size of 64 using FP32 precision.

|  | TIME (MS) | | PEAK MEMORY (MB) | | |
| --- | --- | --- | --- | --- | --- |
| MODELS | TRAINING | INFERENCE | TRAINING | INFERENCE | HARDWARE |
| SIMGNN | 6.9 | 5.9 | 18 | 10 | GPU |
| EGSC | 5.2 | 2.6 | 35 | 25 | GPU |
| GREED | 6.7 | 1.9 | 68 | 62 | GPU |
| ERIC | 17.8 | 1.5 | 24 | 11 | GPU |
| GRAPHEDX-XOR | 33.2 | 29.4 | 146 | 39 | GPU |
| GEDAN ($\lambda = 1$) | 106.3 | 67.6 | 397 | 134 | GPU |
| GEDAN ($\lambda = 0$) | 69.5 | 64.3 | 469 | 156 | GPU |
| EUGENE | - | 9222 | - | 512 | CPU |

All models except EUGENE are trained on GPU, while EUGENE runs on CPU; accordingly, its reported memory usage excludes the process-initialization overhead. All methods are evaluated using FP32 precision with a batch size of 64, ensuring a fair comparison. Models that rely on a matrix-based approximation of GED – specifically GEDAN (both supervised and unsupervised variants) and GraphEdX-Xor – exhibit higher training times and peak memory usage due to the need to allocate and maintain these matrices. Among unsupervised methods, GEDAN is generally less memory-intensive than EUGENE, whereas in the supervised setting it is substantially more memory-demanding compared to standard supervised baselines.

Table 18 reports the average number of trainable parameters for each model. Although GEDAN has fewer parameters than ERIC (Zhuo & Tan, 2022), EGSC (Qin et al., 2021), SimGNN (Bai et al., 2019a), and GREED (Ranjan et al., 2022), it is slower in practice. Notably, GraphEdX (Jain et al., 2024) achieves strong performance with few parameters. EUGENE (Bommakanti et al., 2024), being non-neural, has no trainable parameters. For all other models, we tune hyperparameters to maximize performance on the dataset, taking into account the differences in parameter counts.

Table 18: Average trainable parameters used to approximate the GED

| SIMGNN (BAI ET AL., 2019A) | EGSC (QIN ET AL., 2021) | GREED (RANJAN ET AL., 2022) | ERIC (ZHUO & TAN, 2022) | GRAPHEDX (JAIN ET AL., 2024) | GEDAN ($\lambda = 1$) | ($\lambda = 0$) |
| --- | --- | --- | --- | --- | --- | --- |
| 34,894 | 107,316 | 151,553 | 59,156 | 28,535 | 15,306 | 53,182 |

As shown in Fig. 22, our model is considerably slower than supervised models. This slowdown is primarily due to the overhead introduced by matrix initialization rather than the number of trainable parameters. In contrast, EUGENE (Bommakanti et al., 2024), which is also unsupervised, is significantly slower, taking several seconds to produce a single output. Its CPU-based optimization does not allow for acceleration of the code. Moreover, as illustrated in Figs. 13 and 14, we can train GEDAN for approximately 20 epochs before EUGENE (Bommakanti et al., 2024) completes prediction on the same dataset.

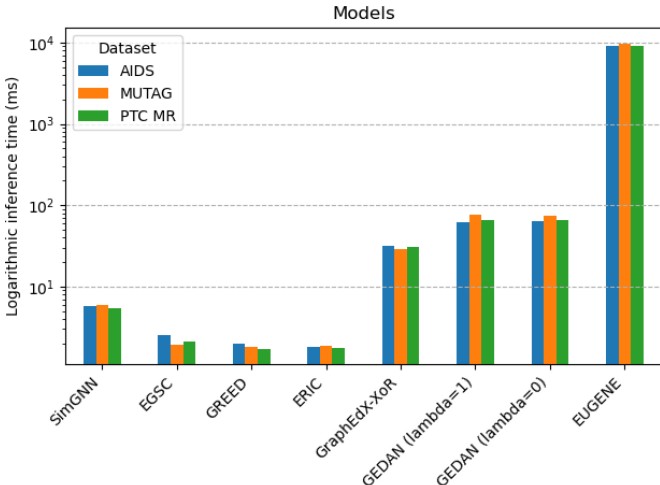

Figure 22: Time comparison in milliseconds on a logarithmic scale of model inference on the GED approximation dataset.

In Fig. 23, we show the average peak memory usage observed during model inference. All models except EUGENE are trained on GPU, while EUGENE runs on CPU; accordingly, its reported memory usage excludes the process-initialization overhead. All methods are trained in FP32 precision with a batch size of 64, ensuring a fair comparison across settings.

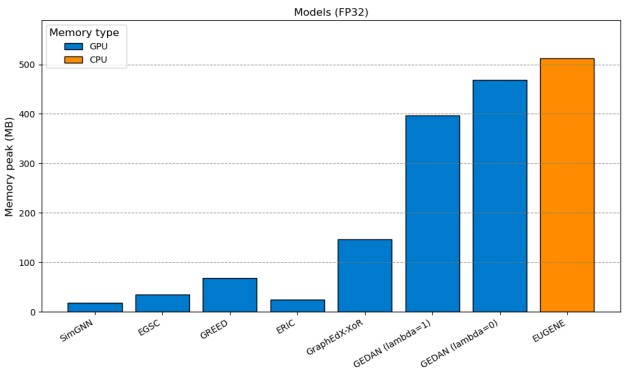

Figure 23: Average peak memory usage during training for all models with FP32 precision and batch size 64.

Models that rely on a matrix-based approximation of GED – specifically GEDAN (both supervised and unsupervised variants) and GraphEdX-Xor – show higher memory consumption because they need to allocate and maintain these matrices. Among unsupervised methods, GEDAN is on average less memory-demanding than EUGENE, whereas in the supervised setting it is significantly more expensive in terms of memory compared to standard supervised baselines.

As mentioned above in the section on limitations, the main bottleneck lies in matrix management. Despite the use of optimized operations such as broadcasting, this component remains the most time-consuming part of the computation.

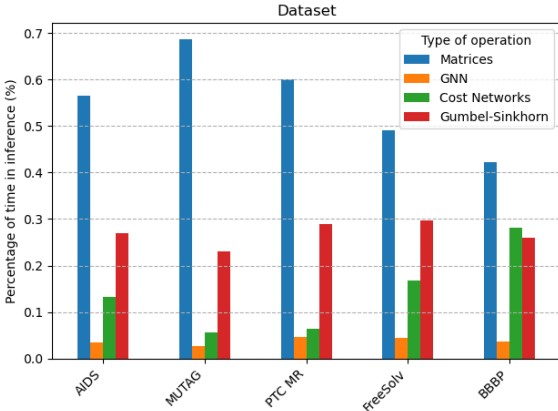

Figure 24: Time comparison of GEDAN's components on inference.

As shown in Fig. 24, matrix operations account for approximately 40% to 70% of the total inference time. The Gumbel-Sinkhorn network constitutes the next most significant contribution, while both the GNN operations and the edit cost computation have a comparatively minor impact.

The inference time for molecular interpretation is compared with other models on a logarithmic scale. In Fig. 3 (a), only GEDAN and GMSM (Pellizzoni et al., 2024) can perform the task, achieving comparable times, with initialization overhead being the main cost. Figures 3 (b) and (c) show inference times for all models. GraphMaskExplainer (Schlichtkrull et al., 2021) and PGExplainer (Luo et al., 2020) include 250 training epochs in their reported times. GATv2 (Brody et al., 2022) is the fastest, as inference only requires visualizing attention weights. Overall, GEDAN is slower than GMSM (Pellizzoni et al., 2024) but faster than GraphMaskExplainer (Schlichtkrull et al., 2021) and PGExplainer (Luo et al., 2020), due to the handling of the maximum cost assigned to node types, which improves local precision at a computational cost.

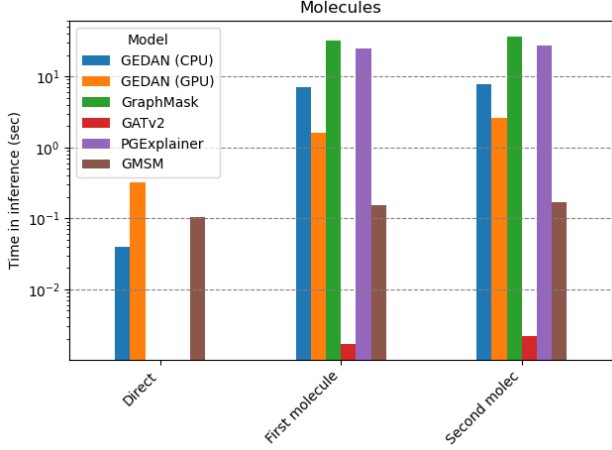

Figure 25: Inference time for molecular interpretation across different models. The direct comparison corresponds to Fig. 3 (a), the first molecule to Fig. 3 (b), and the second molecule to Fig. 3 (c).

## G PREDICTED MOLECULES

In the main paper, we present two representative molecules along with their predictions, highlighting how our model is able to perform both direct comparisons between molecules and consistent analyses of their overall chemical behavior.

Here, we extend this analysis by presenting additional molecular examples to further illustrate these capabilities. Specifically, we study direct molecular comparisons, symmetry-based reasoning, and maintenance of chemical consistency over a larger and more structurally complex set of molecules. These examples serve to demonstrate the robustness and generalizability of our approach, which goes beyond the molecules shown in the main paper.

### G.1 DIRECT COMPARISON

To assess the model's ability to make fine-grained chemical reasoning, we consider pairs of structurally similar molecules with slight variations. These comparisons test whether the model is able to detect and reflect small but chemically significant differences in its predictions. As shown in Fig. 26, the model responds consistently to local changes, capturing many of the expected trends.

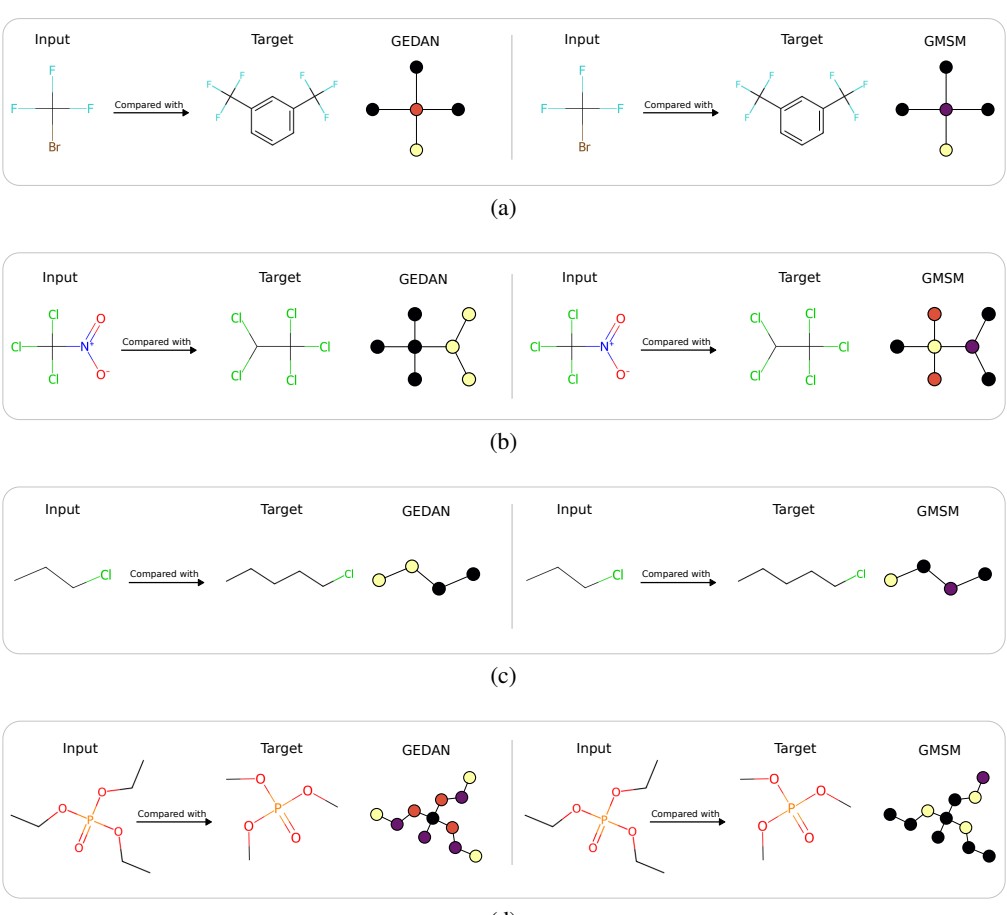

Figure 26: Direct comparison between pairs of similar molecules. Each panel (a – d) shows two molecules differing by a small chemical modification.

### G.2 SYMMETRIC MOLECULES

Molecular symmetry poses a unique challenge for predictive models because it requires invariance or equivariance with respect to permutations of atoms or substructures. The comparisons in which a

single molecule is compared to all others are done one by one. After that, we calculate the average costs at the node level, which gives us a measure of the average transformation cost. This process allows us to determine, on average, which atoms are most important. In Fig. 27, we evaluate the models' ability to produce predictions that respect molecular symmetry, both at the level of global structure and local atoms.

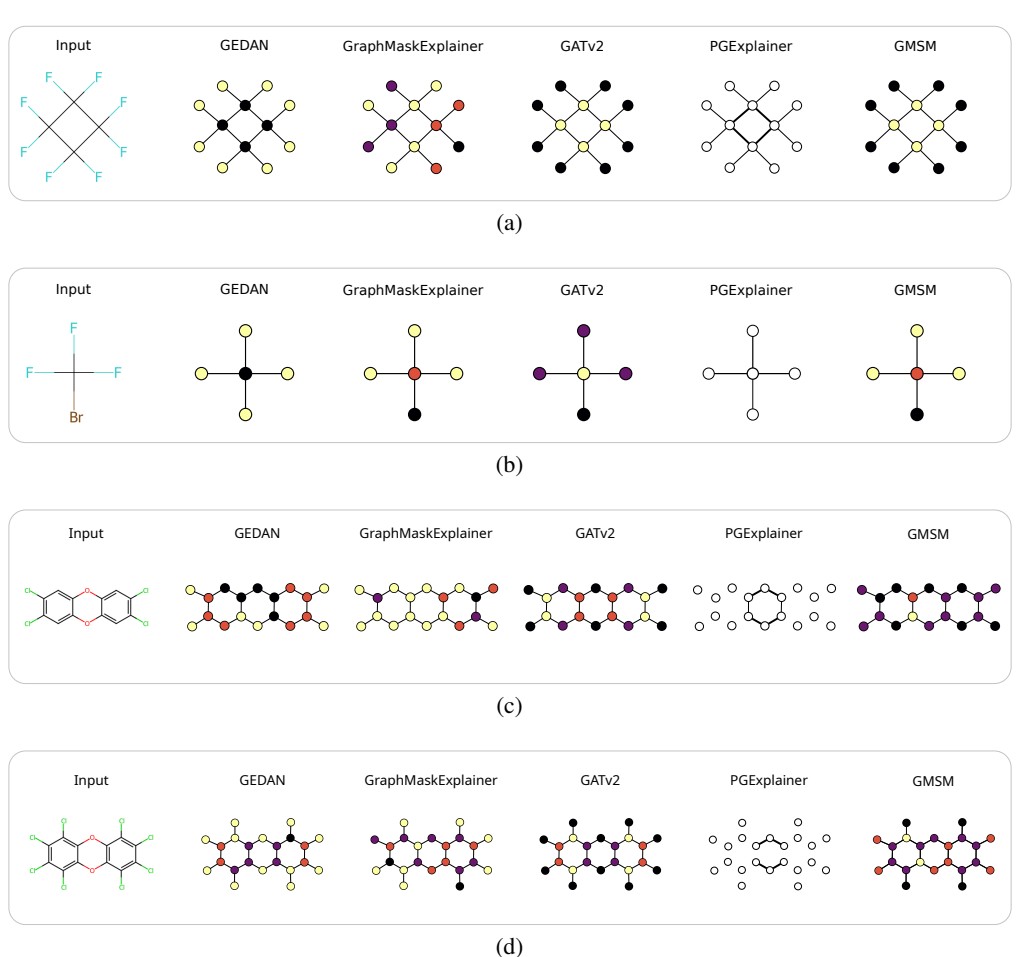

Figure 27: Model predictions on symmetric molecules. Each panel (a – d) illustrates the model's ability to produce equivalent results for symmetrically equivalent atoms or substructures.

## G.3 CHEMICALLY CONSISTENT MOLECULES

To assess the model's ability to generalize to structurally different but chemically similar compounds, we evaluate its predictions on a set of molecules that share some functional properties. In Fig. 28, we complement the main examples by demonstrating that the model preserves chemically meaningful behavior beyond simple or symmetric cases.

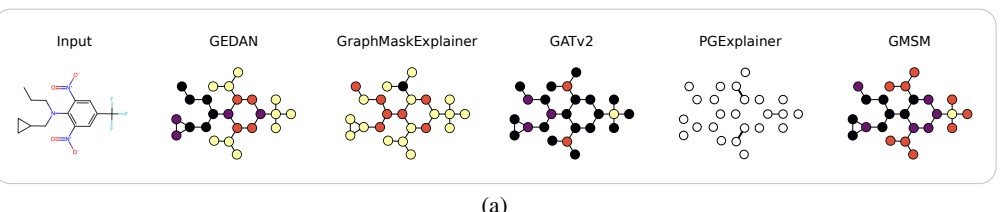

(a)

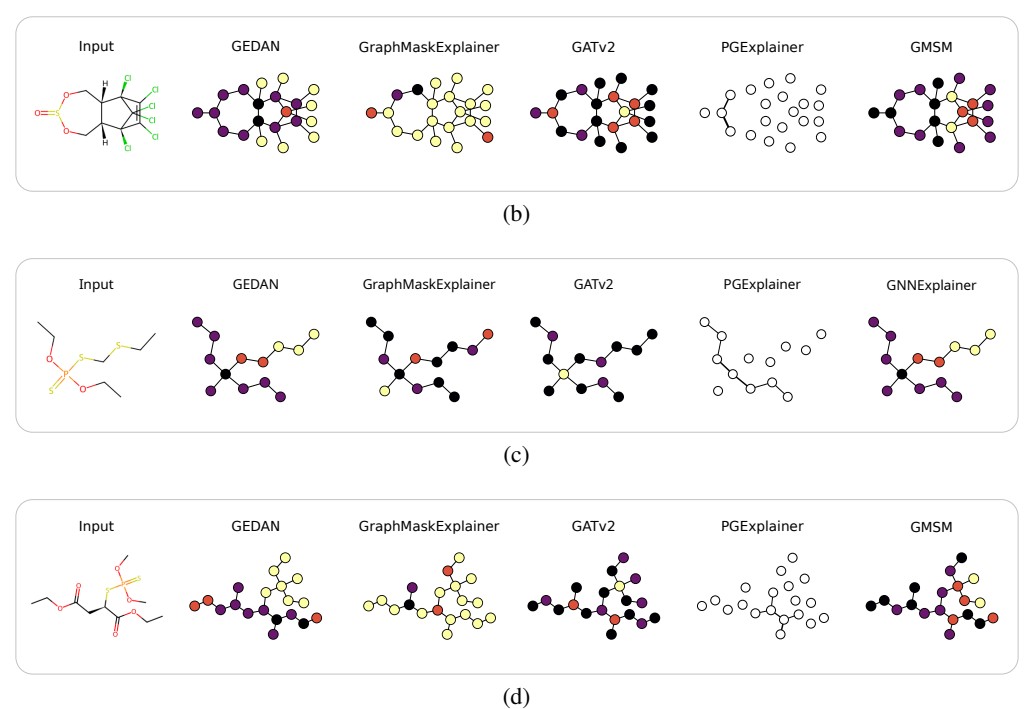

Figure 28: Predicted behavior on chemically consistent molecules. Each panel (a – d) shows a distinct molecule with structurally different motifs but similar chemical properties.

## H SPECIFICATIONS ON THE LIMITATION OF THE NUMBER OF NODES

A major limitation of our method concerns the maximum size that can be handled by the Gumbel-Sinkhorn matrices, which introduce quadratic computational complexity. This aspect limits the model's ability to generalize to large-sized graphs. For this reason, we restricted the application of our approach to small molecules, excluding more complex structures such as peptides or proteins. In addition to the larger size of the graphs, the nature of the problem also changes in these cases, since macro-molecules are strongly affected by three-dimensional folding. Next, we present the maximum size of the graphs that our model can handle and the coverage with respect to the main benchmark datasets currently available online.

Table 19: Size distribution of molecular graphs present in the principal public benchmarks

| | NODES | | | | MAXIMUM SIZE OF GEDAN | | |
|---|---|---|---|---|---|---|---|
| DATASETS | TOTAL | AVG | STD | MAX | 64 | 128 | 256 |
| AQSOL (DWIVEDI ET AL., 2023A) | 7,836 | 15.42 | 8.63 | 156 | 7,812 | 7,834 | 7,836 |
| MD17 (CHRISTENSEN & VON LILIENFELD, 2020) | 576,583 | | | | 576,583 | 576,583 | 576,583 |
| MOLECULENET (WU ET AL., 2017) | 596,815 | | | | 595,116 | 596,659 | 596,764 |
| PCQM4MV2 (HU ET AL., 2021) | 3,378,606 | 14.12 | 2.47 | 20 | 3,378,606 | 3,378,606 | 3,378,606 |
| QM7B (WU ET AL., 2017) | 7211 | 15.42 | 2.69 | 23 | 7,211 | 7,211 | 7,211 |
| QM9 (WU ET AL., 2017) | 130,831 | 18.03 | 2.94 | 29 | 130,831 | 130,831 | 130,831 |
| TUDATASET (MORRIS ET AL., 2020) | 4,789,598 | | | | 4,723,791 | 4,785,518 | 4,789,594 |
| ZINC (GÓMEZ-BOMBARELLI ET AL., 2016) | 220,011 | 23.15 | 4.51 | 38 | 220,011 | 220,011 | 220,011 |
| TOTAL | 9,707,491 | | | | 9,639,961 | 9,703,253 | 9,707,436 |
| TOTAL (%) | | | | | 99.30% | 99.96% | **100.00%** |

