# OpenReview forum: "GEDAN: Learning the Edit Costs for Graph Edit Distance"
_ICLR.cc/2026/Conference — Submitted to ICLR 2026_

### Official Review · Reviewer_iLkT · 2025-10-19

**Soundness:** 3
**Presentation:** 2
**Contribution:** 3
**Rating:** 6
**Confidence:** 3

**Summary:**

This paper addresses the Graph Edit Distance (GED) problem, which is a classical but computationally intractable method for measuring graph dissimilarity. The authors argue that existing neural approximations assume fixed or unit edit costs, which fail to reflect meaningful, task-specific similarities (e.g., in molecular graphs where small topological changes can drastically affect function).

To overcome this, they propose GEDAN, a fully end-to-end differentiable GNN framework that jointly learns node/edge edit costs while approximating GED.

**Strengths:**

1. The paper’s key innovation lies in learning fine-grained edit costs rather than assuming fixed values. This bridges structural similarity and semantic/functional similarity.
2. This paper uses the following neural network to solve the problem: GNNs, GAMs, and Gumbel-Sinkhorn. In detail, GIN-based message passing for expressive node embeddings, Generalized Additive Models (GAMs) for interpretability and modular cost composition, and Gumbel-Sinkhorn networks for differentiable matching demonstrates an elegant synthesis of modern deep-graph and probabilistic-optimization techniques.
3. Extensive experimental results: this paper contains lots of experiments, such as GED approximation quality, Effect of learned vs. fixed edit costs on downstream molecular prediction etc.

**Weaknesses:**

1. The main limitation is that this paper cannot scale up. The experiments shown in this paper mainly focus on graphs with <= 128 nodes.
2. The paper largely focuses on empirical and architectural innovation. A deeper theoretical analysis of learned costs, e.g., proving monotonicity of learned GED w.r.t. functional distance (Eq. 1) or bounding approximation error, would strengthen the contribution.
3. The multi-component training (contrastive + supervised losses, multiple matrices, pre-training of Gumbel-Sinkhorn) may make optimization unstable or sensitive to hyperparameters.

**Questions:**

See Weaknesses

---

> ### Author Response · Authors · 2025-11-26
>
> We thank the reviewer for the feedback and the suggestions.
>
> W1) We agree that our method currently scales up to 128 nodes. However, this covers approximately 99.96% of publicly available small molecule datasets as shown in Appendix H. Scalability is a common challenge for all GED methods, which are generally based on heuristics with at least quadratic complexity, as in the case of GEDAN. Supervised approaches, lacking access to true GED distances for large graphs, also need to rely on approximate GED computations, often introducing upper/lower bounds that affect training. In contrast, GEDAN can operate in zero-shot mode (Table 4), producing competitive predictions without training or GED distances. The inference times reported in Table 17 for GEDAN ($\lambda=1$) reflect the real cost for any graph of that size. For supervised methods, faster inference ignores the overhead of distance computation and model training.
>
> W2) We agree that additional theoretical guarantees would strengthen the work. We discuss this direction in the paper as future work. Extending theoretical results to general contexts is challenging, since guarantees are often context-specific and cost relationships can be highly nonlinear -- for example, a GED lower bound cannot be directly extended to cost edit prediction in molecular applications, as chemically similar molecules may have different structures. For the current work, we added an empirical evaluation of the monotonic alignment between learned GED and functional distance in Appendix E.3. Using Kendall’s $\tau$ and Spearman’s $\rho$ across datasets and cost functions (Table 16), we show that high values indicate that learned edit costs preserve the ordering induced by a known functional distance. The scatter plots in Figure 19 also provide a visual correspondence between the predicted and real functional distances.
>
>
> W3) Our model contains multiple components and requires careful parametrization for optimal performance. The appendices illustrate sensitivity to key parameters. While some parameters have limited impact, pretraining the Gumbel-Sinkhorn module is crucial (Table 3). Without this pre-training, the model no longer works correctly in unsupervised or zero-shot mode, while in supervised mode, it introduces significant noise and reduces performance.

---

### Official Review · Reviewer_hDhH · 2025-10-28

**Soundness:** 3
**Presentation:** 3
**Contribution:** 2
**Rating:** 4
**Confidence:** 2

**Summary:**

This paper provides a novel method to optimize the edit costs in the graph edit distance, in order to align it with the functional dissimilarity between graphs.

**Strengths:**

This paper addresses a difficult topic, which is the estimation of the edit costs in GED for a given task. The proposed method relies on recent advances in the literature, such as Jain et al. (2024), and provides recent advances by exploring the Gumbel-Sinkhorn that allows approximation of permutation matrices.

The appendices provide essential information to better understand the proposed method, with extensive experimental analysis.

**Weaknesses:**

The title of the paper is too general. There are many other methods that learn the edit costs for graph edit distance. The authors are missing some of the related literature on this topic.

The proposed method relies on a pre-trained Gumbel-Sinkhorn network to approximate the Linear Sum Assignment Problem (LSAP). While this is an interesting approach, it is also a major weakness because one needs to pre-train the network. This leads to several issues, as described in the following.

The resulting method depends on the pre-training and its biases, such as the used method which preferentially selects node pair representations with minimum distance. Other choices would lead to different biases.

The pre-training relies on a pre-defined size (e.g. 64 x 64) and therefore is very restricted once the size is fixed. Padding dummy nodes to get to the same size, as proposed in the paper, is not a clever strategy, as it highly increased the computational complexity. Moreover, binary masks are used at the end, also increasing the computational cost.

The computational complexity of the method is a major weakness. There is room for improvement.

The appendices provide essential information to better understand the proposed method and the conducted choices. However, it turns out that some of the choices are not well justified or explained in the main body. For instance, it turns out that the authors are not using the original Gumbel-Sinkhorn network, but a modified version. It would have been relevant to clearly specify this in the main paper and motivate this choice, including an ablation study.

**Questions:**

No further questions.

---

> ### Author Response · Authors · 2025-11-26
>
> Thank you for your feedback and comments.
>
> We have expanded the related work section by adding two of the earliest statistical and probabilistic approaches (Neuhaus & Bunke, 2004; 2007), as well as non-fully differentiable methods (Cortés et al., 2019; Garcia-Hernandez et al., 2020; Rica et al., 2021), along with the most recent differentiable works already cited (Pellizzoni et al., 2024; Heo et al., 2024). The number of existing methods for learning edit costs remains limited, mainly because GED is NP-hard and, in deep learning, has mostly been treated as a measure to be approximated. However, learning edit costs requires a fully differentiable and unsupervised approach, since the optimal edit costs are not known a priori.
>
>
> For this reason, our method builds upon the LSAP reformulation of (Riesen & Bunke, 2009), which approximates GED via bipartite graphs with an original cost of $O(N^3)$. Using bipartite graphs requires padding; although not ideal, it is necessary for the LSAP-based approximation. As shown in Table 4, our approach works in a zero-shot context and can handle up to 128 nodes (Figure 16), covering approximately 99.96% of the small molecule datasets available online (Appendix H), a result that is difficult to achieve with other methods. We also added Appendix F with a more detailed computational efficiency analysis. Scaling the method and reducing the cost-matrix complexity is our active line of work, and we agree that there is room for improvement. Currently, the method has a complexity of $O(LKN^2)$, where $L$ is the number of Sinkhorn iterations and $K$ is the depth of the GNN on the graph, with $L$ and $K$ much smaller than $N$.
>
>
> The modifications to Gumbel-Sinkhorn are reported in the appendix. We use standard modifications already used in the literature -- Truncated Gumbel to reduce gradient variance and an early stopping criterion for Sinkhorn -- which allow us to approximate the LSAP value with fewer iterations. The ablation study shows the necessity of pre-training, which -- as you pointed out -- introduces bias by favoring certain pairs. This arises from the LSAP formulation itself, where candidate pairs are preselected to reduce an NP-hard problem to $O(N^3)$. Our goal is to better penalize mismatched pairs, yielding a more accurate GED estimate and a more reliable matching, crucial for precise edit-cost computation.

---

### Official Review · Reviewer_nDHo · 2025-10-31

**Soundness:** 3
**Presentation:** 3
**Contribution:** 3
**Rating:** 6
**Confidence:** 4

**Summary:**

The paper proposes GEDAN, a GNN framework for approximating the Graph Edit Distance (GED). The main contribution is a novel, end-to-end differentiable method for learning fine-grained, context-aware edit costs. This learned cost function aims to align the traditional topological GED metric with task-specific functional similarity, which is often misaligned in real-world data like molecules. The model uses a GIN for multi-scale node embeddings, formulates GED as a matching problem solved via a pre-trained Gumbel-Sinkhorn network, and introduces an auxiliary MLP to learn the edit costs. The paper presents an unsupervised mode to approximate fixed-cost GED and a supervised mode to learn costs for downstream functional tasks.

**Strengths:**

(1) The paper addresses a well-known and significant limitation of standard GED: the mismatch between topological distance (using fixed/unit costs) and functional similarity, particularly in domains like computational chemistry.
(2) The core idea of building an end-to-end, differentiable framework to learn these costs is novel and valuable, as it allows GED to be optimized directly for downstream tasks.
(3) The model offers a path to interpretability by analyzing the learned edit costs.

**Weaknesses:**

(1) Hard limit of 128 nodes due to Gumbel-Sinkhorn complexity is critically restrictive.
(2) Table 3 shows RMSE of 2509.84 (No-PT) vs 3.54 (PT) - the method completely fails without careful initialization. This is reasonable, but is a Gumbel–Sinkhorn trained on random matrices optimal for real GNN-induced cost matrices?
(3) Domain generality not fully shown. All downstream is molecular / chem. That is fine, but GED gets used in scene graphs, program graphs, point-cloud graphs, document structure graphs, knowledge graphs, etc. Even a small “OGB-small” or “program-AST” style experiment would make the “task-aligned edit costs” story more general.

**Questions:**

(1) Could you report wall-clock time and peak GPU memory for various methods tested?
(2) Why GAM specifically? Could you show that a simpler learned scalar-weighted sum performs worse?
(3) You mention 128 as the practical cap. Suppose I have 300-node program graphs or scene graphs — what is your recommendation approach?

---

> ### Author Response · Authors · 2025-11-26
>
> We thank the reviewer for your feedback and comments.
>
> W1) We agree on the dimensionality limitations introduced by Gumbel-Sinkhorn, which is an active area of research in our group as we work towards scaling the model to larger graphs. Despite this constraint, our model can be applied to 99.96% of online benchmark datasets on small molecules (Appendix H). Other GED methods are often less scalable: supervised approaches require ground-truth distances (NP-hard) or heuristics with complexity often greater than $O(N^2)$, in addition to subsequent training. Our method, with complexity $O(LKN^2)$, where $L$ is the number of Sinkhorn iterations and $K$ is the depth of the GNN on the graph, allows for distance prediction even in a zero-shot setting (Table 3). The computational complexity mainly arises from the LSAP method (Riesen & Bunke, 2009) we rely on, which has an original complexity of $O(N^3)$.
>
>
> W2) Gumbel-Sinkhorn is trained on many matrices to quickly approximate LSAP with a complexity of $O(L*N^2)$. We agree that, while effective, it may not be the optimal strategy.
>
>
> W3) We agree that other types of graphs could benefit from a more general story. We choose molecular graphs due to their relevance and the difficulty of estimating edit costs, even for experts, as well as the extensive literature that facilitates result interpretation. To demonstrate the broader applicability of our approach, in Appendix E.3 we added an experiment on an artificial dataset modeling a GPU node network, where the objective is to estimate the total cost of transformation between networks. This case is more interpretable than program graphs or knowledge graphs, which will be considered in future work.
>
>
> Q1) We have updated Appendix F (Computational Efficiency Analysis), showing both inference and training time and peak memory usage. This allows a better positioning of GEDAN relative to the literature, highlighting that the model is more expensive to train but relatively fast at prediction.
>
> Q2) The GAM is chosen for its interpretability and ability to express more complex functions. In Appendix E.3, we added an experiment showing how different types of cost functions affect performance (Table 16). In simple scenarios, a scalar-weighted sum is valid, but less effective in non-linear problems, such as the GPU node network.
>
> Q3) For large graphs (e.g., program graphs or scene graphs with ~300 nodes), GED approximation requires heuristics with complexity comparable to GEDAN. We recommend, when possible, reducing the graph size to allow direct application of the framework, accepting a resolution trade-off. The alternative would be to use very large matrices, which may lead to suboptimal performance. We are also exploring reduction methods that minimize information loss, while recognizing that they are not always applicable or generalizable to all graph types.

---

### Official Review · Reviewer_n6VT · 2025-11-01

**Soundness:** 2
**Presentation:** 2
**Contribution:** 2
**Rating:** 2
**Confidence:** 4

**Summary:**

This paper proposes to learn context-aware edit costs for Graph Edit Distance (GED) computation. The method builds on GNN-based node representations and incorporates a pre-trained Gumbel–Sinkhorn network to approximate graph node correspondences, enabling node-specific, context-dependent edit cost estimation rather than relying on fixed uniform costs.

**Strengths:**

- The paper clearly identifies an important gap: neural GED models typically assume fixed scalar edit costs, while many real applications require context-specific edit penalties.  Highlighting this limitation and attempting to address it is meaningful and timely. Learnable edit costs tailored to the downstream graph relevance task is still an open an interesting challenge.


- Usage of a pre-trained Gumbel–Sinkhorn network to approximate permutations is an interesting choice. If this pretraining meaningfully improves matching quality or reduces search complexity, this could be a useful contribution for the broader GED and graph matching community.

**Weaknesses:**

- The paper frames its goal as learning the classical edit operation costs for GED — specifically the node/edge insertion and deletion costs $a^{\oplus}, a^{\ominus}, b^{\oplus}, b^{\ominus}$. However, these costs appear to remain fixed in the proposed architecture. Instead, the model learns a node-to-node pairwise substitution cost $c_{i,j}$. Instead of  learning edit costs,  in the traditional GED sense this is rather learning affinity measure (aka edit costs)  for soft matching. Thus the central problem remains unaddressed.

- Given that  the learned parameters serve as node-pair affinity measures feeding into a differentiable assignment solver, the method becomes closely related to prior neural graph matching frameworks. I do not see anything being done differently  beyond the use of a *pre-trained* Gumbel–Sinkhorn network.


- The choice of cosine distance to measure node pair affinity is not well justified. Cosine normalization removes information contained in the embedding norms, which can be important for distinguishing neighborhoods. For example, consider two graphs with identical node labels: a 4-cycle and a 4-clique. When using normalized cosine similarity, their node embeddings become indistinguishable, because the difference in neighborhood cardinality is reflected primarily in the magnitude of aggregated features rather than their direction.

- Notation issues:    Key symbols   $a^{\oplus}, a^{\ominus}, b^{\oplus}, b^{\ominus}$ are introduced abruptly and not formally defined. Section 2.4 in particular lacks precise mathematical exposition. As of now the details regarding the loss function are extremely vague.

- The paper uses AIDS,MUTAG, etc. with size filtering. This method has been previously seen to result in structural leakage [1] across datasets. The authors should ensure and clarify their  aprroach towards detecting/managing structurally isomorphic graphs.

[1] Position: Graph Matching Systems Deserve Better Benchmarks, ICML 2024

**Questions:**

Questions are mainly related to the aforementioned weaknesses.

- Are the classical edit operation costs actually learned? $a^{\oplus}, a^{\ominus}, b^{\oplus}, b^{\ominus}$ appear to remain fixed during training, while the model instead learns a node–node substitution cost \(c_{i,j}\).

-  The high-level pipeline of learning node affinities and using differentiable matching, seems similar to prior work. What is the conceptual difference ?

- Justification for cosine distance?

- Was structural isomorphism check done during dataset creation?

---

> ### Author Response · Authors · 2025-11-26
>
> We thank the reviewer for your feedback and comments.
>
> Q1) We appreciate the opportunity to clarify this point. GEDAN is able to learn both the classical scalar edit costs $a^{\oplus}, a^{\ominus}, b^{\oplus}, b^{\ominus}$ and the substitution costs $c_{i,j}$. To prevent ambiguity, we updated Section 2.4 by adding the following sentence:
>
> >In the supervised setting, where target labels are provided, the model leverages both the edit costs produced by the functions $f_{\theta}^{(k)}$ and the scalar parameters $a^{\ominus}, a^{\oplus}, b^{\ominus}, b^{\oplus}$ to align GED with the target labels.
>
> In the unsupervised setting, both the outputs of $f_{\theta}^{(k)}$ and the scalar costs must remain fixed, as allowing them to vary, the model could exploit a shortcut to minimize itself by setting all edit costs to zero, failing to learn the graph topology (Formula 2, Appendix A).
>
> Relying on the LSAP formulation (Riesen \& Bunke, 2009), substitution costs implicitly encode insertions and deletions through $f_{\theta}^{(k)}(\epsilon,j)$ and $f_{\theta}^{(k)}(i,\epsilon)$. For completeness, we added experiments in Appendix E.3 examining the trajectories of the learned edit costs. These experiments show, for instance, that despite identical random initialization of the scalar terms, the model converges to the expected insertion/deletion ratios (MUTAG configurations 4 and 5, Figure 17). Moreover, $f_{\theta}^{(0)}(\epsilon,j)$ converges to the expected value on datasets with known insertion costs for specific nodes (Figure 18). We believe that this can provide a simple but direct evidence that GEDAN learns both generic and fine-grained edit costs in an end-to-end way.
>
>
>
> Q2) Our method is directly grounded in the LSAP formulation (Riesen & Bunke, 2009), enabling the estimation of GED itself rather than only a node matching. We introduce a fine-grained matching component absent from the classical formulation and render the entire pipeline differentiable, allowing for both zero-shot GED approximation and self-minimizing training. To prevent trivial solutions, we impose constraints and employ a pretrained Gumbel-Sinkhorn network. Such a procedure is not applicable to standard matching-based methods, which typically require task-specific training and do not yield the GED directly.
>
> While GMSM (Pellizzoni et al., 2024) uses matchings to learn edit costs, it is not designed to estimate GED nor to operate in an unsupervised setting. Our framework unifies these capabilities without architectural modifications, supporting both supervised and unsupervised training. To our knowledge, no existing approach achieves this level of flexibility while remaining competitive on two separate tasks -- GED estimation (Table 1) and edit-cost learning (Table 2) -- and while providing localized interpretability (Figure 3).
>
>
> Q3) Cosine distance is normalized to prevent degenerate minimization via norm collapse in the unsupervised setting. The observation regarding structures such as a 4-cycle versus a 4-clique with identical node labels is correct: $\mathbf{M}_k$ alone cannot distinguish such neighborhoods. To address this, we incorporate the matrix $\mathbf{D}$, which encodes node degrees (Formula 5). Matrix $\mathbf{D}$ contributes to the calculation of the permutation (Formula 6), the calculation of GED (Formula 7), and the learning of edit costs (Formula 8). Consequently, even when $\mathbf{M}_k$ exhibits limited discriminative power, the model can still identify and penalize structural differences, as the two graphs exhibit different degree patterns preserved in $\mathbf{D}$.
>
>
> Q4) We agree that the treatment of isomorphic graphs is essential. In the original submission (lines 1255--1256) we mentioned this issue but did not describe the procedure in detail. We updated Section D.1 to include the suggested reference and explicitly state that isomorphic graphs are removed by computing GED between graph pairs and retaining GED $=0$ only along the diagonal of the GED matrices. Additionally, we added Figure 9, which reports the GED matrices for configurations 1 (uniform), 2 (symmetric), and 4 (asymmetric), illustrating the distribution of zero entries. Although we did not perform a statistical analysis dedicated to this condition, all models are trained on identical splits, ensuring fair performance comparisons.
>
>
> Notation issues) The notation $a^{\oplus}, a^{\ominus}, b^{\oplus}, b^{\ominus}$ is formally introduced in Appendix B to maintain readability in the main text. Loss definitions are likewise detailed in the appendices: Appendix C.3 provides the loss for the self-minimizing model, and Appendix E.2 describes the losses used for contrastive learning for both regression (FreeSolv) and classification (BBBP). Sections 2.2 and 2.4 are updated with direct references to these appendices to assist readers in finding this information.

---

### Meta-Review · Area_Chair_de1e · 2026-01-02

**Summary:**

The paper proposes GEDAN, an end-to-end differentiable method for learning fine-grained, context-aware edit costs. This learned cost function aims to align the traditional topological GED metric with task-specific functional similarity, which is often misaligned in real-world data like molecules.

While the efficacy of the method is competitive against considered baselines, it has several outstanding concerns that prohibits me from accepting this work. The key concerns include:

* There is a hard limit of 128 nodes due to Gumbel-Sinkhorn complexity is critically restrictive. The rebuttal could not address this limitation.
* The inference time of the proposed techniques is 10 to 50 times slower than baselines that it marginally outperform in some datasets. Overall, this dramatically slower inference times keeps the contribution of this work marginal at best.
* Several state of the art baselines have not been compared to and/or cited. Some of them are listed below. In addition, non-neural baselines have been ignored (https://github.com/dbblumenthal/gedlib).
  - GRAIL: Graph Edit Distance and Node Alignment Using LLM-Generated Code, ICML 25
  - Computing Graph Edit Distance via Neural Graph Matching, VLDB 23
  - D. B. Blumenthal, N. Boria, J. Gamper, S. Bougleux, and L. Brun. “Comparing heuristics for graph edit distance computation”, VLDB J. 29(1), pp. 419-458, 2020, https://doi.org/10.1007/s00778-019-00544-1
* The approach fails to establish domain generalizability, a capability already showcased by more recent methods such as Grail.

**Reviewer Concerns:**

**Concerns addressed by the rebuttal**

* Clarification of edit-cost learning: Authors explained that GEDAN learns both classical scalar edit costs and substitution costs, grounding this in the LSAP formulation and providing additional experiments.

* Cosine distance justification: They acknowledged its limitations but argued that degree matrices supplement discriminative power, preventing collapse in unsupervised settings.

* Treatment of isomorphic graphs: The rebuttal added details on how isomorphic graphs were removed and clarified dataset handling.

* Notation and loss definitions: Authors pointed to expanded appendices and updated sections to improve clarity.

The key outstanding concerns are discussed above in the "Summary"

**Reviewer Scores:**

Based on the rebuttal, the anticipated final scores from the reviewers are as follows:

* Reviewer n6VT has an initial rating of 2. No comments were received when the discussion was frozen. The rebuttal does address several of the concerns and I expect the score to increase to 4 or 6.

* Reviewer nDHo has an initial rating of 6 and mentions the limitation to 128 nodes as a critical limitation. The rebuttal did not address this. So I do not see scope of the rating improving. In light of the outstanding critical concern, a rating of 4 is also possible.

* Reviewer hDhH shares the same concern wrt scalability and has a rating of 4. I find the rebuttal to this reviewer weak. Hence, the rating is unlikely to improve further.

* Reviewer iLkT has a rating of 6, and shares the same concern wrt scalability. I do not see the rating improving following the rebuttal.

Overall, there are critical limitations, and based on the rebuttal I do not see any of the reviewers championing this work

---

### Decision · Program_Chairs · 2026-01-26

Reject